# Dynamic molecular network analysis of iPSC-Purkinje cells differentiation delineates roles of ISG15 in SCA1 at the earliest stage

Hidenori Homma [1,6], Yuki Yoshioka [1,6], Kyota Fujita [1,2,6], Shinichi Shirai [3], Yuka Hama[3], Hajime Komano[4], Yuko Saito[5], Ichiro Yabe [3], Hideyuki Okano [4], Hidenao Sasaki[3], Hikari Tanaka [1] ✉ & Hitoshi Okazawa [1] ✉

Better understanding of the earliest molecular pathologies of all neurodegenerative diseases is expected to improve human therapeutics. We investigated the earliest molecular pathology of spinocerebellar ataxia type 1 (SCA1), a rare familial neurodegenerative disease that primarily induces death and dysfunction of cerebellum Purkinje cells. Extensive prior studies have identified involvement of transcription or RNA-splicing factors in the molecular pathology of SCA1. However, the regulatory network of SCA1 pathology, especially central regulators of the earliest developmental stages and inflammatory events, remains incompletely understood. Here, we elucidated the earliest developmental pathology of SCA1 using originally developed dynamic molecular network analyses of sequentially acquired RNA-seq data during differentiation of SCA1 patient-derived induced pluripotent stem cells (iPSCs) to Purkinje cells. Dynamic molecular network analysis implicated histone genes and cytokine-relevant immune response genes at the earliest stages of development, and revealed relevance of ISG15 to the following degradation and accumulation of mutant ataxin-1 in Purkinje cells of SCA1 model mice and human patients.

The focus of neurodegenerative disease research is increasingly shifting to earlier stages because neurodegeneration seems to be triggered much earlier than expected. For instance, the recent failure of anti-amyloid antibodies in patients with pre-onset familial Alzheimer's disease (AD)[1] suggests that the pathology begins before extracellular amyloid aggregation. Consistently, multiple groups independently reported pre-aggregation pathologies of AD[2–5] that are highly homologous. Although the recent success of lecanemab in clinical trials of patients with mild cognitive impairment (MCI) or early-stage AD is striking[6], its effect on cognitive symptoms is smaller than expected. In addition, the exceptional effect of lecanemab, which targets protofibrils before aggregation, supports the significance of pre-aggregation

pathology. It is theoretically possible to prevent or ameliorate the pathologies of familial neurodegenerative diseases caused by monogenic mutations using a genetic approach in human patients. For this to become a reality, it is essential to determine when the earliest molecular pathology occurs and what it entails.

Spinocerebellar ataxia type 1 (SCA1) is a familial neurodegenerative disease that mainly affects cerebellar Purkinje cells and spinal motor neurons. Genetic analyses revealed that CAG repeat expansion in an exon of the *Ataxin-1* (*Atxn1*) gene is linked to the onset of SCA1[7]. As observed in other polyglutamine (polyQ) diseases including Huntington's disease, the mutant gene produces abnormal Atxn-1 proteins whose full-length or processed

---

[1]Department of Neuropathology, Medical Research Institute, Tokyo Medical and Dental University, 1-5-45, Yushima, Bunkyo-ku, Tokyo 113-8510, Japan. [2]Research Center for Child Mental Development, Kanazawa University, 13-1 Takaramachi, Kanazawa-shi, Ishikawa 920-8640, Japan. [3]Department of Neurology, Faculty of Medicine, Graduate School of Medicine, Hokkaido University, Kita 15, Nishi 7, Kita-ku, Sapporo 060-8638, Japan. [4]Department of Physiology, Keio University School of Medicine, 35 Shinanomachi, Shinjuku-ku, Tokyo 160-8582, Japan. [5]Department of Neuropathology, Tokyo Metropolitan Institute of Gerontology, 35-2 Sakae-cho, Itabashi-ku, Tokyo 173-0015, Japan. [6]These authors contributed equally: Hidenori Homma, Yuki Yoshioka, Kyota Fujita. ✉e-mail: tanaka.npat@mri.tmd.ac.jp; okazawa.npat@mri.tmd.ac.jp

forms accumulate and aggregate to generate neuronal inclusion bodies[8]. Although the symptomatic onset of neurodegenerative disease often occurs in adulthood, developmental pathologies occur in many such diseases. A significant body of evidence supports the notion that developmental pathologies contribute to the pathology of SCA1 despite symptomatic onset in middle age[8,9].

The first evidence of developmental pathologies in SCA1 was studies identifying that retinoic acid-related orphan receptor α (RORα) target genes are essential for cerebellar development[10]. Normal Atxn1 forms a ternary complex with Tip60 and RORα, while mutant Atxn1, the disease-causing gene product in SCA1 that possesses an expanded polyQ tract, binds Tip60 but is unable to form the ternary complex with RORα, ultimately decreasing expression of genes essential for cerebellar development[10]. The gene products of Yes-associated protein 1 (YAP) are comprised of two alternatively spliced isoforms (YAP1-1/YAP1 or YAP1-2/YAP2) and also regulate the ability of Atxn1 and RORα[11] to form the ternary complex. Further, mutant Atxn1 inhibits formation of the ternary complex with RORα, suppressing expression of genes necessary for cerebellum development[11]. Drug-induced transgenic supplementation of YAP during early developmental stages but not during adulthood prevents Purkinje cell neurodegeneration and late-onset cerebellar ataxia and motor dysfunction[11], underscoring the significance of developmental pathology in SCA1.

Atxn1 interacts with multiple transcriptional regulators such as SMRT[12], Gfi-1[13], and Capicua[14,15] via the AXH domain of Ataxin or mediators of RNA splicing such as PQBP1[16], potentially via liquid–liquid phase separation (LLPS) of intrinsically disordered proteins. Due to this complex regulatory network, in SCA1, transcriptional and post-transcriptional regulation of gene expression would be disrupted significantly and heterogeneously in neurons, glia, and non-neuronal cells. Orr and colleagues determined the effect of Axtn1 disruption on gene expression by RNA sequencing (RNA-seq) of brain samples from mutant Atxn1 transgenic (Atxn1-Tg) and mutant Atxn1 knock-in (Atxn1-KI) mice at 5–28 weeks of age[17]. However, pathological gene network changes during developmental pathologies of SCA1 remain unknown, despite the potential impact of these events on the course of later pathological processes[11]. In a previous report, induced pluripotent stem cells (iPSCs) were generated from SCA1 patients and differentiated to pan-neurons[18]. Expression levels of several genes were measured by quantitative PCR[18]. However, the earliest developmental pathology of SCA1 during differentiation to Purkinje cells remains unknown.

In the present study, we performed comprehensive RNA-seq analyses of iPSCs generated from SCA1 patients and evaluated all gene expression profiles during differentiation from totipotent stem cells to Purkinje cells. We used an originally developed analytical approach to elucidate the cause-and-effect relationship underlying temporal changes of molecular networks, Induction Method-based Analysis for Dynamic molecular networks (iMAD). iMAD-iPSC analyses revealed the dynamics of SCA1 molecular networks originating from histones[19–22] in iPSCs (temporal upstream) to cytokines in immune responses including interferon (IFN)-stimulated gene 15 (ISG15)[23,24] in Purkinje cells (temporal downstream). This pathological regulatory network could initiate brain inflammation reported in later stages of SCA1[25] and could be an early trigger for accumulation and aggregation of diseased protein in advanced SCA1[26–28]. Consistently, we identified that ISG15 was upregulated in the plasma of human patients and suggest that ISG15 is a potential biomarker for SCA1.

## Results
### Generation of iPSCs from SCA1 patients
We generated iPSCs from lymphocytes derived from two SCA1 patients by infecting isolated lymphocytes with retroviral vectors expressing the four transcription factors of pluripotency (Oct3/4, Sox2, Klf4, and c-Myc)[29–31]. Patient A carried 25/44 repeats, and patient B carried 29/41 repeats, as demonstrated by automated fragment analysis of lymphocytes, and patient A has 26/46 repeats in peripheral blood and 26/46 and 26/47 in cell lines, and patient B 29/43 in peripheral blood and 29/43 and 29/44, 29/46, 29/47 in derived cell lines, as demonstrated by Sanger sequencing

(Supplementary Fig. 1a). Given that the peak of fragment sizes is manually located in the automated fragment analysis causing occasionally such a minor difference in repeat numbers due to the technical limitation, these results from two methods basically confirmed the corresponding $(CAG)_mCAT(CAG)_n$ sequences. Three iPSC lines generated from each patient carried similar repeat numbers in Atxn1 gene (Supplementary Fig. 1a). The six lines exhibited normal iPSC morphology (Supplementary Fig. 1b) and equivalent levels of pluripotency markers such as Oct3/4, SSEA-4, and Tra1-60 (Supplementary Fig. 1c). Chromosome analysis with G-banding revealed that all SCA1-iPSC lines possessed normal chromosomes (Supplementary Fig. 1d). HKC1-3 and 201B7 iPSC lines purchased from the RIKEN bank were used as normal controls. Morphologies of feeder-free normal iPSCs and SCA1-iPSCs, which were used for RNA-seq in later experiments, were similar (Supplementary Fig. 1b).

### Impaired differentiation of SCA1-iPSCs to pan-neurons and Purkinje cells
We examined abnormalities of SCA1-iPSCs during differentiation (Supplementary Fig. 2), identifying slower proliferation times of feeder-free iPSCs (Supplementary Fig. 2a). Embryonic bodies (EBs) differentiated from SCA1-iPSCs were also transiently smaller than those differentiated from normal iPSCs (Supplementary Fig. 2b, c). Following an established protocol[29–31], we further differentiated normal and SCA1 EBs to pan-neurons, and measured neurite extension length (Supplementary Fig. 2d) and density of PSD95-positive spine-like protrusions as indicators of neuronal differentiation (Supplementary Fig. 2e). Both parameters suggested impaired neuronal differentiation of pan-neurons from SCA1-iPSCs.

We reported previously that mutant Atxn1 interacts with HMGB1 to inhibit its physiological functions in transcription and DNA damage repair[32], and that supplementation of HMGB1 by AAV-HMGB1 delivery alleviates late-stage SCA1 pathology of mutant Atxn1-KI mice[33]. We thus determined if AAV-mediated expression of HMGB1 would improve the phenotypes of iPSC-derived pan-neurons (Supplementary Fig. 2d, e). Consistent with our previous findings in late-stage SCA1 pathology, HMGB1 recovered the developmental pathologies of impaired neurite extension (Supplementary Fig. 2d) and decreased dendritic spines (Supplementary Fig. 2e), supporting the therapeutic effect of HMGB1 on SCA1 developmental pathology.

Subsequently, we examined abnormalities of SCA1-iPSCs following differentiation to Purkinje cells (Supplementary Fig. 3) and identified that multiple aspects of differentiation were disrupted (Fig. 1). As shown in representative images of Purkinje cells differentiated from SCA1 and normal iPSCs, the number of Purkinje cells (defined as Calbindin+ cells) was not decreased in Purkinje cells differentiated from SCA1-iPSCs (Fig. 1), while neurite lengths of Calbindin+ Purkinje cells were decreased in cells differentiated from SCA1-iPSCs (Fig. 1). The results concerning Purkinje cells and neurite length were confirmed by quantitative analyses (Fig. 1).

### Static molecular network analysis of SCA1-iPSCs during differentiation to Purkinje cells
We prepared RNA samples from normal and SCA1 iPSC lines (201B7, HKC-1, and HKC-3 for control cell lines and SCA1A-26, SCA1B-8, and SCA1B-37 for patient cell lines), as well as from EBs and Purkinje cells differentiated from the iPSC lines, and performed RNA-seq data analysis to compare normal and SCA1 cells at each differentiation stage (Supplementary Fig. 4a, b). We searched for differentially expressed genes (DEGs) by DESeq2[34] at iPSC, EB and Purkinje cell stages (Supplementary Data 1) and revealed a relationship between fold change (SCA1-iPSC per normal iPSC) and reliability ($p$-value) as shown in the volcano plot, in which statistically significant genes ($|$ $Log_2$ Fold Change $| > 0.5$ by DESeq2[34]) are indicated in green (decreased in SCA1-iPSCs) or red (increased in SCA1-iPSCs) (Fig. 2a). The DEGs (Supplementary Data 1) were subjected to KEGG-based pathway analysis (Fig. 2b), elucidating the functional pathways of differentially expressed genes. Fisher's exact test indicated that 11, 24 and 11 pathways were significantly affected (adjusted $p$-value < 0.001) at iPSC,

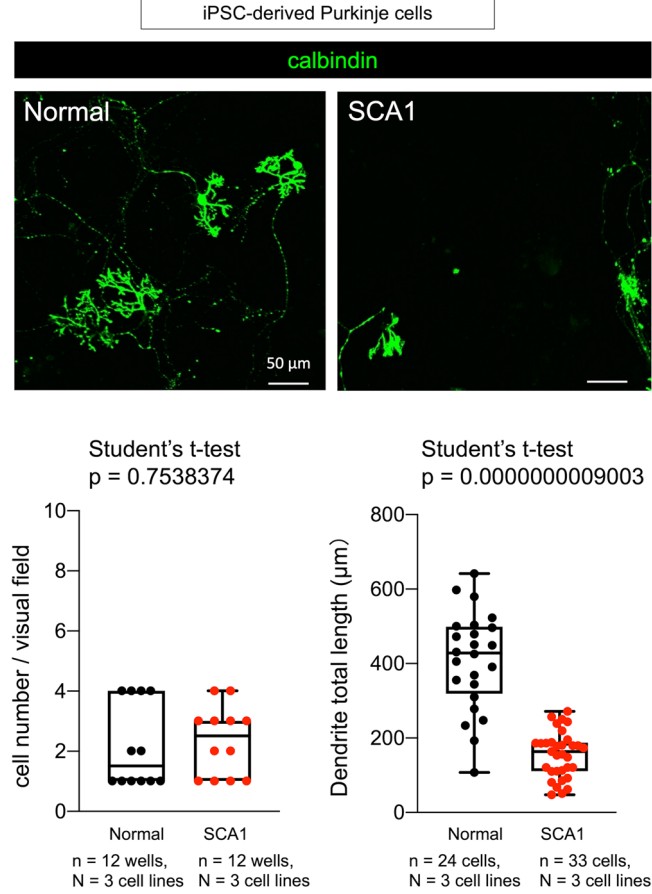

**Fig. 1 | Characterization of SCA1-iPSC-derived Purkinje cells.** Immunocytochemistry of iPSC-derived Purkinje cells with Calbindin, the specific marker of Purkinje cells. The lower left graph compares mature Purkinje cell numbers and the lower right graph compares Purkinje cell neurite lengths between cells differentiated from SCA1-iPSCs and normal iPSCs. Eight visual fields from two independent wells were observed. The two-tailed Student's *t*-test was used for statistical examination. The box plot shows the median and 25–75th percentile, and whiskers represent data outside the 25–75th percentile range.

EB and Purkinje cell stage, respectively (Fig. 2a, right lists). Interestingly, multiple cytokine-related pathways (cytokine-cytokine receptor interaction; neuroactive ligand-receptor interaction; viral protein interaction with cytokine and cytokine receptor) were most significantly affected throughout from iPSC to Purkinje cell stages (Fig. 2a, right lists).

We have previously shown that HMGB1 and YAP play critical roles in developmental and post-developmental pathology of SCA1[11,32,33]. Though these genes were not selected in the statistic network analysis, further analyses of RNA-seq read numbers of each gene revealed that expression levels of YAP and its partner transcription factor TEAD1, but not the TEAD1 cofactor TAZ, were decreased in SCA1-iPSCs at the EB stage (Fig. 2c). Intriguingly, expression patterns of the YAP and TEAD1 were similar during iPSC differentiation (Fig. 2c), suggesting the relevance to the retarded growth of SCA1-iPSC and SCA1-EB (Supplementary Fig. 2a–c) given that TEAD1-YAP is the critical transcriptional factor complex regulating cell proliferation[35,36].

**Dynamic molecular network analysis of SCA1-iPSCs during differentiation to Purkinje cells**

The static molecular network analysis clearly revealed the functional impact of the Atxn1 gene mutation at each stage of iPSC differentiation. However, the static molecular networks cannot explain cause-result relationships from one time point to the next. When we tried to connect the selected nodes at two sequential time points based on protein–protein interaction

(PPI) databases, many nodes were disconnected, and such disconnected nodes might not influence the later development of pathology (Supplementary Fig. 5). In addition, the statistic network analysis might neglect critical genes contributing the pathology. Thus, we developed an original mathematical method, induction Method-based Analysis for Dynamic molecular networks (iMAD), to elucidate the cause-effect relationship underlying chronologically dynamic molecular network changes (Fig. 3). A cause-result relationship was not taken into account when generating the static molecular network, but was the basis of generating the dynamic molecular network (Fig. 3a) (see details in the figure legend and Methods).

We selected potentially significant genes at specific time points not by changes in expression level at the time point but by their potential effects on relevant genes at the next time point (Fig. 3a). To do this, we employed PPI databases to identify relevant genes (see details in Methods). All genes connected to the gene of interest via a single edge were considered to be relevant. The degree of impact was calculated as the percentage of differentially expressed genes among the relevant gene set at the next time point, which was statistically compared with the percentage of all changed genes in RNA-seq with a Fisher's exact test (Fig. 3a). We selected genes that had a high reliability (*p* < 0.05 in Fisher's exact test with post-hoc Benjamini-Hochberg procedure) (Fig. 3b, Supplementary Data 2) as significant. We colored the node red (Fig. 3b) and drew edges (orange lines) from red nodes to relevant genes changed at the next time point (Fig. 3b). Unexpectedly, the dynamic molecular networks evaluated by two distinct criteria converged to a network comprised of a small number of key regulatory genes (core nodes) from the iPSC stage to the Purkinje cell stage (Fig. 3b).

**Functional connections of core molecules in dynamic network**

The convergence of a dynamic molecular network of SCA-iPSCs during differentiation setting the selection condition at less than 5% of FDR, could reflect a specific cellular function impaired in SCA1 pathology (Supplementary Fig. 6). Core nodes were converged to ten genes at the iPSC stage, which included albumin (*ALB*), histone H2B type-1F (*HIST1H2BF*), histone H2B type-1H (*HIST1H2BH*), histone H4 (*HIST4H4*), Glutathione S-transferase Mu 1 (*GSTM1*), Serine protease 1 (*PRSS1*), Dehydrogenase/reductase SDR family member 2 (*DHRS2*), Gamma-aminobutyric acid receptor subunit beta-2 (*GABRB2*), Cyclin A1 (*CCNA1*) and TYRO protein tyrosine kinase-binding protein (*TYROBP*) (Fig. 4a). Mapping the converged genes to a PPI network (String ver.11.5 URL: https://string-db.org/) revealed the impact of each of the core genes at iPSC stage on DEGs at Purkinje cell stage (Fig. 4b). It was surprising that three histone genes (*HIST1H2BF, HIST1H2BH, HIST4H4*) at iPSC stage possessed large impacts on gene expression changes at Purkinje cell stage (Fig. 4b, Supplementary Data 3).

Gene ontology (GO) analysis of connected genes at Purkinje cell stage revealed a high frequency inclusion of cytokine-relevant genes regulating immune responses in *HIST1H2BF*-originated, *HIST4H4*-originated and *TYROBP*-originated networks (Fig. 4c, Supplementary Data 4), in consistence with the result from statistical network analyses (Fig. 2b). These data suggested that the initial pathological change, i.e., abnormal expression of histone genes led to disordered expression profiles of cytokine-relevant genes at developmental stages far earlier than previously expected. The second finding of interest was that ten core genes at iPSC stage were redundantly included in the pathway from other nodes at Purkinje cell stage or at EB stage, suggesting that the core nodes compose feed-back or feed-forward loops (Fig. 4d), which suggested a relative significance of *HIST1H2BF* among ten core genes. In addition, three pathways originated from *ALB, CCNA1* and *PRSS1* included *ATXN1* in the network at Purkinje cell stage (Supplementary Data 3).

Therefore, we focused on cytokine-relevant genes in the dynamic network originated from *HIST1H2BF*, and found two secretary cytokines (*ISG15* and *VEGFA*) included in the list of connected nodes at Purkinje cell stage while most of the other genes in the list were cell membrane molecules (Supplementary Data 4). The relevance of VEGF to SCA1 was reported previously[37] where the authors described reduction of *VEGFA* mRNA in *ATXN1* transgenic mouse model[17] at 3 months of age while they did not

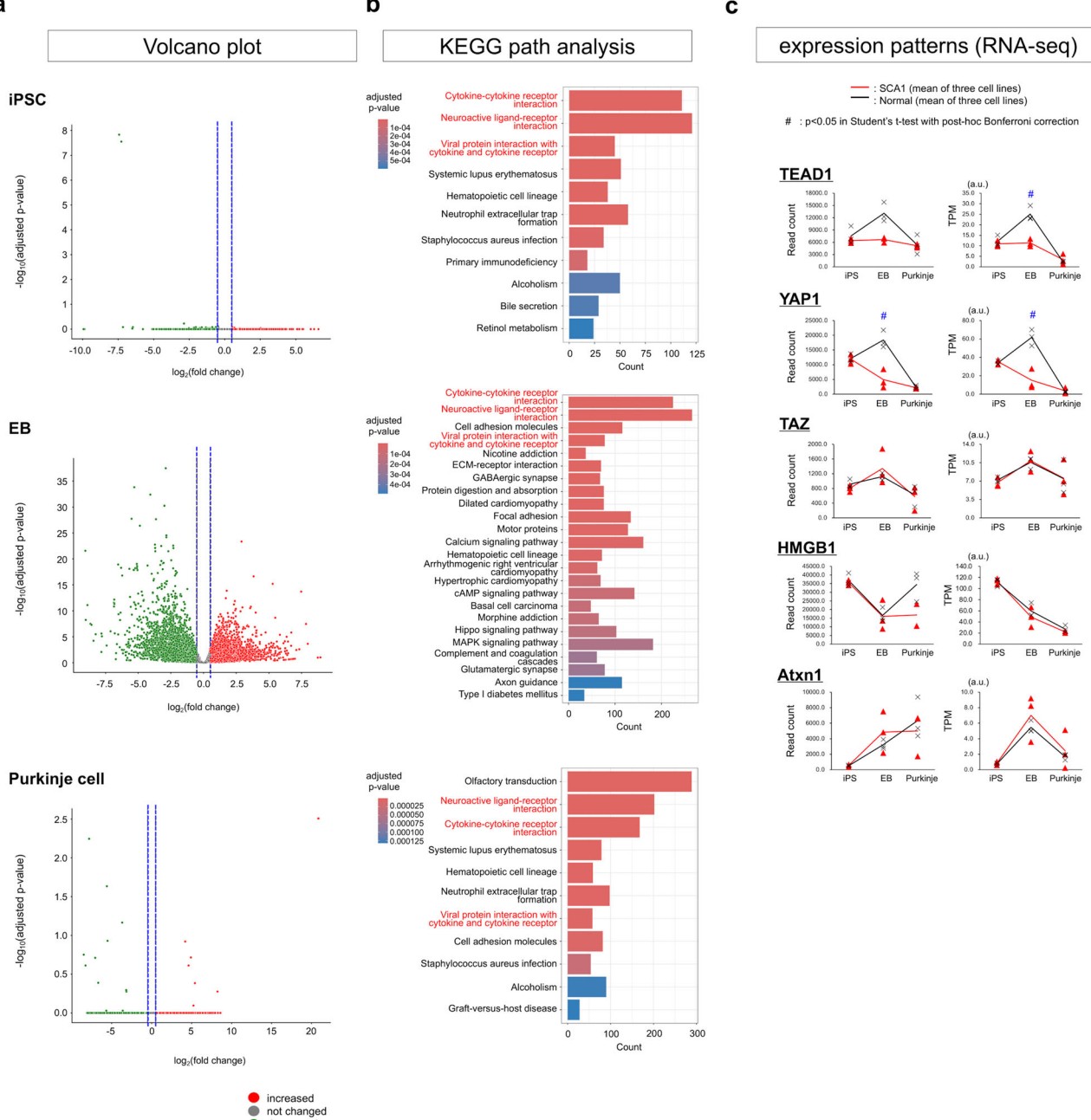

**Fig. 2 | Gene expression profile analysis of iPSCs, EBs, and Purkinje cells.**
**a** Graphs show volcano analysis of RNA-seq data obtained from iPSCs, EBs, and Purkinje cells. Fold changes of mRNA expression of SCA1-derived cells in comparison to normal cells (x-axis) were calculated as $\log_2$ (mRNA expression in SCA1 cells / mRNA expression in normal cells). Adjusted $p$-values evaluating the difference between mRNA expression of SCA1-derived cells and normal cells were calculated by Welch's $t$-test ($N = 3$) and expressed as $\log_{10}$ ($p$-value) along the y-axis. Numbers of DEGs were 1161, 8752 and 245 at iPSC, EB and Purkinje cell stage, respectively. **b** Panels show KEGG pathway analysis of the changed genes (Results of

pathways with the top 15 smallest $p$-values are shown). Differentially expressed genes were plotted on a KEGG pathway, and the number and ratio of changed genes were calculated in each pathway. Bar colors indicate $p$-values of Fisher's exact test with post-hoc BH procedure. **c** Chronological mRNA expression patterns of key molecules with known roles in SCA1 (TEAD1, YAP, TAZ and HMGB1). The two-tailed Student's $t$-test with the post-hoc Bonferroni correction was used for multiple comparisons. $P = 0.029$ in the right panel of TEAD1 at the EB stage, $P = 0.019$ in the left panel of YAP1 at the EB stage, and $P = 0.016$ in the right panel of YAP1 at the EB stage.

confirm the change in human patients. In our analysis, VEGFA was increased at PC stage differentiated from human iPSC (Supplementary Data 4). The discrepancy might be due to different developmental stages or different species between their and our studies. *ISG15* was increased ($\log_2$ Fold Change > 0.5 by DESeq2) at EB and Purkinje cell stages in the pathway from HIST1H2BF. ISG15, as one of the interferon-stimulated genes, could be in the downstream of interferon (IFN) and JAK-STAT pathways included by NK cell mediated cytotoxicity pathway. Though ISG15 was

implicated in Ataxia-Telangiectasia (A-T)[38], it remains unknown whether and how ISG15 contributes to the SCA1 pathology. Therefore, we decided to focus on the pathway from *HIST1H2BF* to *ISG15* in this study.

**Meta-analysis using public databases supports the iPSC dynamic network**
To verify the dynamic molecular network derived from our RNA-seq data of iPSCs, we searched for publicly available RNA-seq databases related to

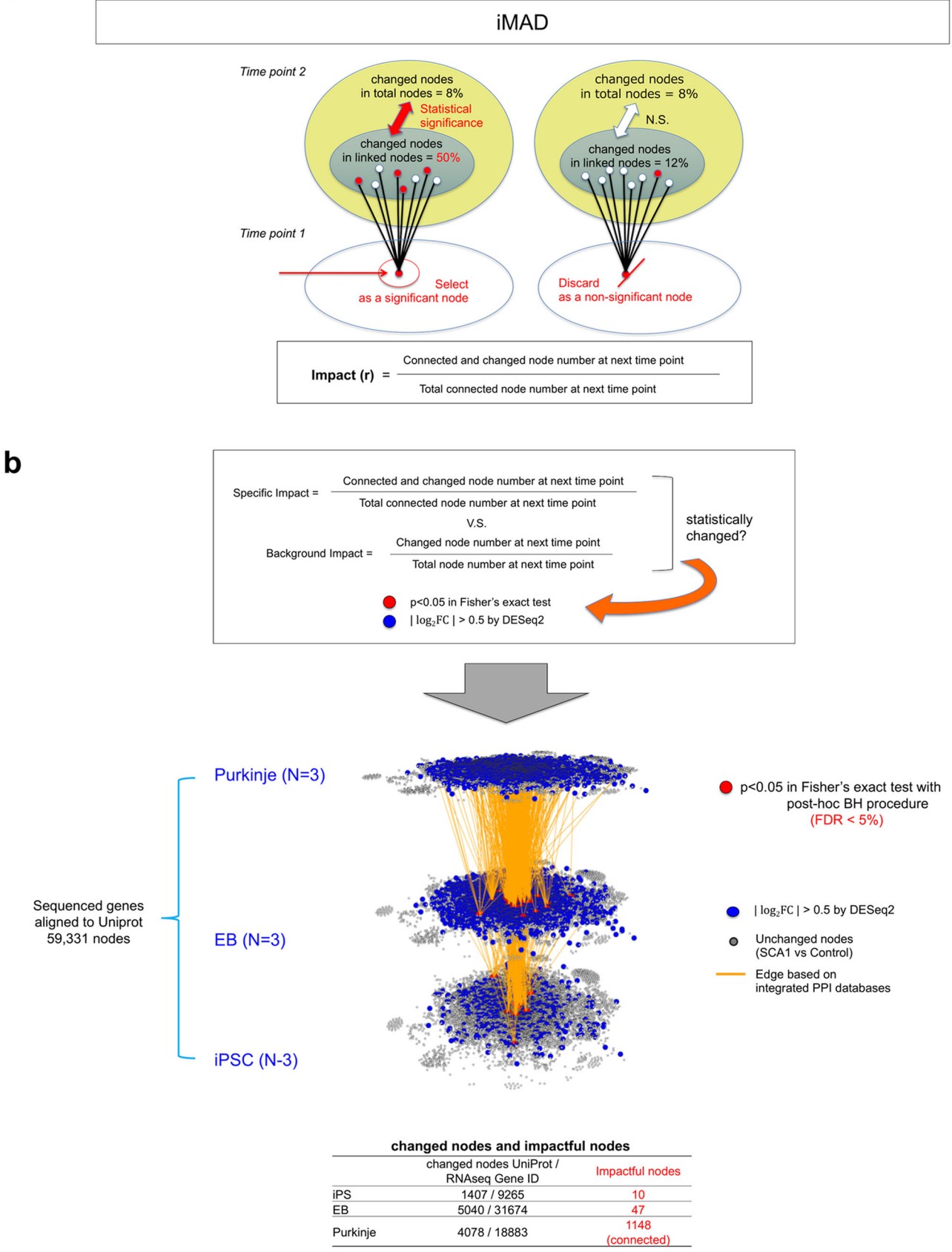

**Fig. 3 | Prediction of core chronological pathway by dynamic molecular network analysis with iMAD. a** Concept of iMAD. Impact of a gene at time point 1 on the connected genes at time point 2 was evaluated, based on the assumption that the change of an impactful gene would influence relevant genes of a significantly larger number than the background change of all genes at the following time point. The left

image is representative of an impactful gene, and the right image is representative of a nonimpactful gene. **b** DEGs were selected by | $Log_2FC$ | > 0.5 by DESeq2, and impactful genes were selected by iMAD with statistical significance ($p < 0.05$ in Fisher's exact test with post-hoc BH procedure: FDR < 5%).

SCA1. As far as we could find, RNA-seq databases are available for the SCA1 mouse model but not for human iPSCs, iPSC-derived neurons, iPSC-derived Purkinje cells, or human post-mortem brains. Therefore, we performed iMAD analysis using the public databases of two SCA1 mouse models, Sca1[154Q/2Q] knock-in mice (*Atxn1*-KI mice)[39] and Pcp2-*ATXN1*-82Q/2Q transgenic mice (*Atxn1*-Tg mice)[40], and examined how the mouse dynamic molecular network is connected with our iPSC dynamic molecular network (Fig. 5, Supplementary Fig. 6).

The mouse age that corresponds to the Purkinje cell stage of iPSCs could not be rationally determined; therefore, we first connected or inserted the network data of the iPSC-Purkinje cell stage to multiple mouse ages and examined the frequency of connected nodes (Supplementary Fig. 6). Unexpectedly, nodes at the iPSC-Purkinje cell stage were more frequently connected to nodes at aged stages in both *Atxn1*-KI and *Atxn1*-Tg mice (Supplementary Fig. 6), suggesting that iPSC-Purkinje cells correspond to aged Purkinje cells in vivo.

Therefore, we connected the dynamic network of iPSCs to that of mouse models in two ways. First, the Purkinje stage was connected to the 5 week stage (connected network). Second, the 28 week stage was connected to the Purkinje stage (inserted network). We tested whether the two networks included the ISG15-related network (Fig. 5). As expected, both networks included the cytokine-relevant sub-networks based on gene ontology originating from HIST1H2BF at the iPSC stage (Fig. 5). The details of nodes in these networks were listed (Supplementary Data 5). Similarly, connected networks of iPSCs to the *Atxn1*-KI or *Atxn1*-Tg mouse network included the ISG15-relevant sub-network (Supplementary Fig. 7). Collectively, these results verified iMAD as a method for dynamic molecular network analysis and indicated that the cytokine-relevant sub-network is a key pathway.

## ISG15 is a core molecule at late developmental stages of SCA1

Collectively, these findings prompted us to use a SCA1 mouse model, mutant heterozygous *Atxn1*-KI (Sca1[154Q/2Q]) mice[39], to investigate the roles of ISG15 at late developmental or childhood stages of disease pathology, a much earlier stage than symptomatic onset at middle age. To determine the potential connection of ISG15 with adult SCA1 pathology, we stained cerebellar sections of normal and mutant Atxn1-KI mice at 4, 13, and 56 weeks of age for ISG15. As expected, ISG15 staining in Calbindin[+] Purkinje cells was higher in transgenic mice than in age-matched wild-type controls (Fig. 6a). Further, Purkinje cell ISG15 signal was increased in transgenic mice as early as 4 weeks of age (Fig. 6a). ISG15 staining was also modestly increased in granule cells and neuropils of the molecular layer (Fig. 6a). Interestingly, ISG15 signal was highest at 4 weeks of age in *Atxn1*-KI mice, and the difference between control and transgenic mice subsequently decreased (Fig. 6a, b). We also performed western blot analysis for ISG15 in whole cerebellum lysates from normal and mutant Atxn1-KI mice at 4, 9, and 32 weeks of age (Fig. 6c). Quantitative analyses of the results further supported the postnatal increase of ISG15 in Purkinje cells and in the cerebellum in *Atxn1*-KI mice (Fig. 6c). Given that IFNs upregulate transcriptional gene expression of ISGs[41] and that upregulation of IFNβ was observed in another type of cerebellar degeneration, Ataxia-Telangiectasia (A-T)[38], we examined developmental changes of the IFNβ protein level and STAT1 phosphorylation at Tyr701 (Fig. 6d). As expected, IFNβ upregulation and downstream STAT1 activation preceded upregulation of ISG15 in the cerebellum (Fig. 6e). Analyses of ARIH1 and UBE2L6 expression patterns (Fig. 6f) revealed that increased expression of ARIH1 (ISGylation E3 ligase), rather than UBE2L6 (ISGylation E2 enzyme), contributed to increased ISGylation.

## ISGylation impairs protein degradation of Atxn1

Consistently, the whole-gel images of ISG15 western blots of whole cerebella from *Atxn1*-KI mice and their normal siblings revealed increases of several ISG15-conjugated proteins in *Atxn1*-KI mice, with the most affected proteins at 40 and 100 kDa (Fig. 7a, left panel). Interestingly, a previous study reported that two ISG15 molecules are conjugated with a single ubiquitin (Ub) molecule, forming a 40 kDa protein and delaying protein degradation by the Ub-proteasome system[38]. The hybrid heterotrimer comprised of one Ub molecule and two ISG15 molecules was also chemically synthesized[42], but the biological meaning of the hybrid target protein modification remains unknown. We suspected that the 40 kDa band observed in our western blot (Fig. 7a, red arrow) might be the heterotrimer. Moreover, we suspected that the ISG15-conjugated protein at 100 kDa (Fig. 7a, blue arrow) could potentially be Atxn1 because the sum of their molecular weights is about 100 kDa and that ISGylation of Ub could influence degradation of ubiquitylated mutant Atxn1. Reprobing of the same filter indicated that the 100 kDa band corresponded to ISG15-conjugated Atxn1, but it remained uncertain whether the 40 kDa band corresponded to the heterotrimer of ISG15 and Ub (Fig. 7a, right panels).

To test our hypotheses regarding ISGylated proteins, we reprobed the western blot filter sequentially with anti-ISG15, anti-Ub, and anti-Atxn1 antibodies (Fig. 7b). Anti-Ub and anti-Atxn1 antibodies labeled both the 100 and 40 kDa bands. Therefore, we assume that Atxn1 is modified by both Ub and ISG15 (Fig. 7c). Moreover, we performed immunoprecipitation of cerebellar tissue samples from *Atxn1*-KI mice and their normal siblings at 4 weeks of age with an anti-ISG15 antibody and performed western blotting with an anti-Ub or anti-Atxn1 antibody (Fig. 7d). The experiments supported that the 100 kDa protein was Atxn1 conjugated with both Ub and ISG15 (Fig. 7d, thin blue arrow). In addition, immunoprecipitation of cerebellar tissue samples from *Atxn1*-KI mice, but not their normal siblings, detected both ubiquitinated and ISGylated Atxn1 at higher molecular weights (Fig. 7d, thick blue arrow and blue arrowhead). The high molecular weight bands could be Atxn1 molecules cross-linked by transglutaminase 5[43] that escaped from protein degradation. Immunostaining of Atxn1 and ISG15 or Ub and ISG15 revealed they co-localized in the nucleus (Fig. 7e), indicating that Ub conjugated with ISG15 modifies mutant Atxn1 proteins.

Furthermore, we expressed Myc-tagged normal/mutant Atxn1 protein in HeLa cells and performed cycloheximide pulse-chase analysis to examine whether ISGylation influences degradation of Atxn1 protein. Pre-treatment with siRNA-ISG15 to knockdown ISG15 accelerated degradation of mutant Atxn1 in comparison with pre-treatment with siRNA-control (Fig. 7f, g). On the other hand, pre-treatment with siRNA-ISG15 did not affect degradation of FLAG-Ku70, which was used as a control, or normal Atxn1 (Fig. 7f, g). Moreover, inclusion of MG132 in the medium during the pulse-chase period suppressed degradation of mutant Atxn1 protein (Fig. 7f, g). In contrast with the effect of the proteasome inhibitor MG132, the autophagy inhibitor bafilomycin A did not influence the result, demonstrating that autophagy-dependent protein degradation is not directly related to this process (Fig. 7f, g). These results collectively indicated that enhanced ISGylation inhibited Ub-proteasome-dependent degradation of mutant Atxn1 in Purkinje cells in the context of SCA1 pathology.

## ISG15 is a surrogate marker for early stage of human SCA1

Next, we examined ISG15 in human SCA1 pathology by using post-mortem patient brains and peripheral blood of living patients. We included three patients and controls whose information is described in the Methods. ISG15 staining in Calbindin[+] Purkinje cells was greater in SCA1 patients than in non-neurological disease human controls (Fig. 8a, b). In Purkinje

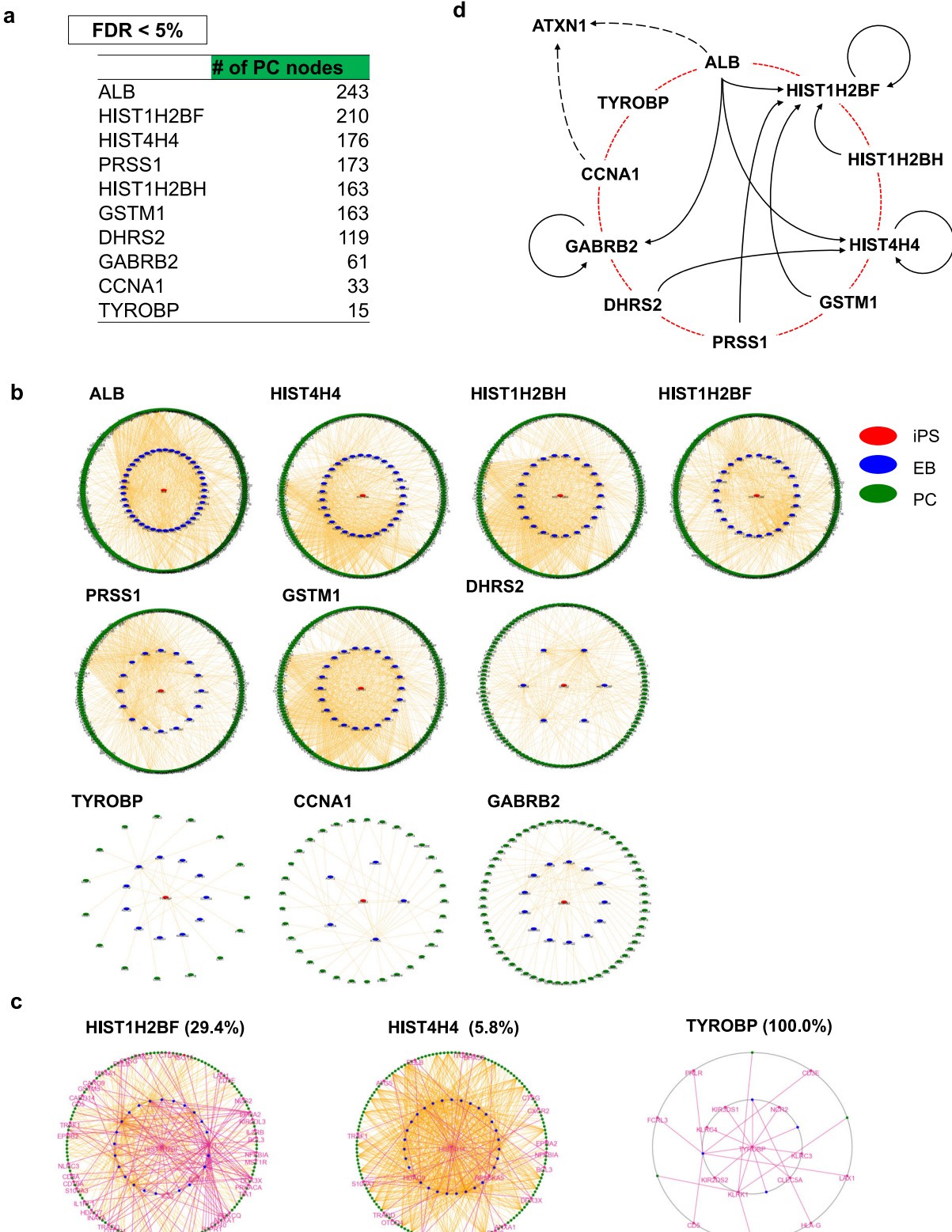

**Fig. 4 | Prediction of chronological SCA1 pathology by impactful genes.**
**a** Impactful genes selected by criteria (FDR < 5%) at the iPSC stage and their target DEG numbers at the PC stage connected by PPI database (Target). **b** Molecular networks composed of impactful genes at the iPSC stage (red) and EB stage (blue) selected by iMAD and of target genes (PPI-based connected DEGs) at Purkinje cell stage (green). **c** High frequencies of involvement of GO-based cytokine-relevant sub-networks (purple edges) in three core networks originated from iPSC stage (orange edges). In TYROBP all edges were related to cytokine. Cytokine-relevant edges / total edges in iMAD (purple lines/orange lines) of each networks are indicated in percentages. **d** Digraph analysis illustrating feed-back or feed-forward loops among ten core genes.

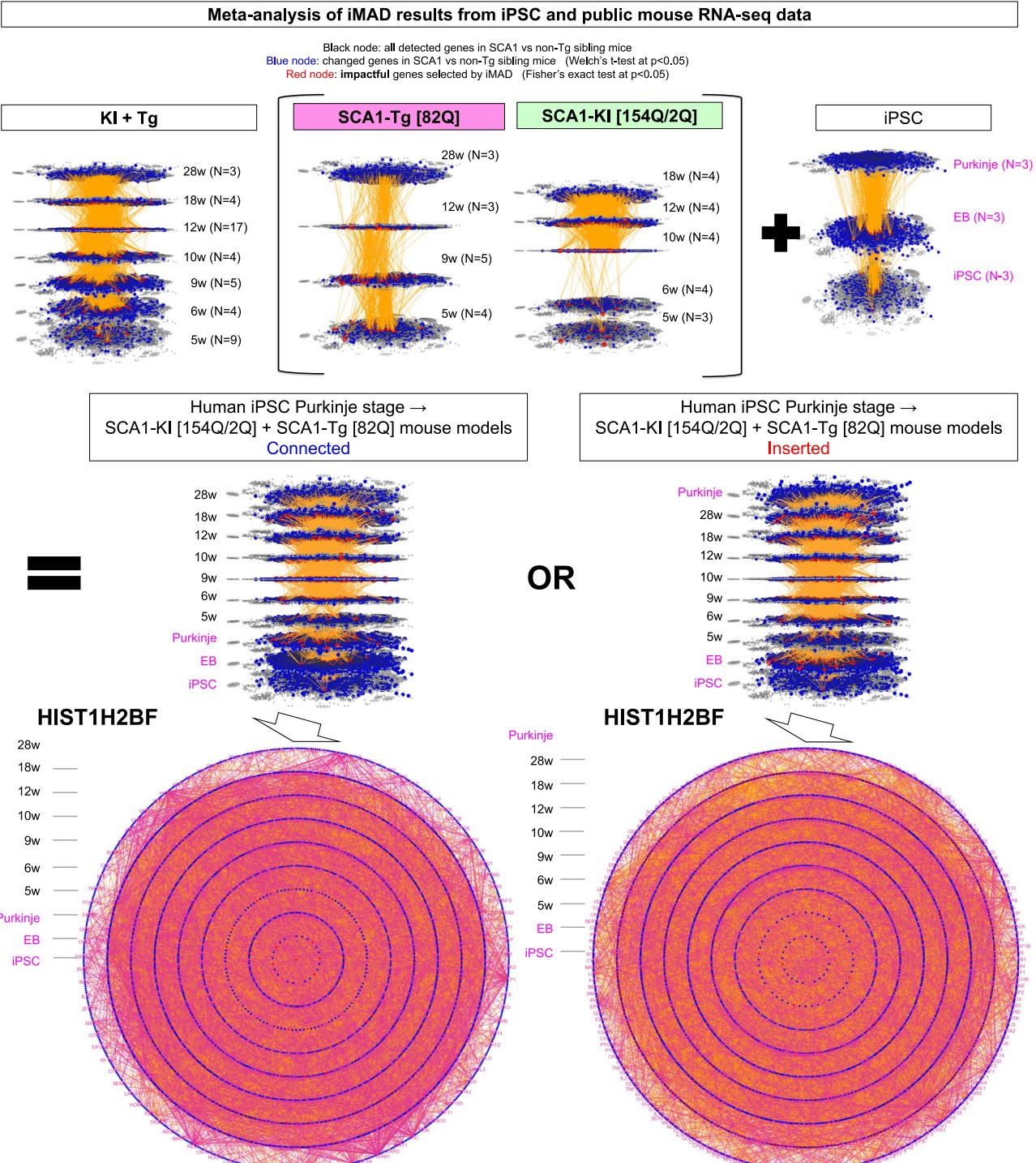

**Fig. 5 | Meta-analysis with public databases supports the iPSC dynamic network.** A dynamic molecular network was generated by iMAD analysis using publicly available RNA-seq data of cerebellar tissues from Atxn1-KI or Atxn1-Tg mice, and then connected with the dynamic molecular network of iPSCs (Fig. 4c) (upper row). Before the connection, the similarity of the connection characteristics of the Atxn1-KI and Atxn1-Tg mouse networks with the iPSC network was preliminarily examined (Supplementary Fig. 8). The connected dynamic network from human iPSCs to cerebellar tissues of the two SCA1 mouse models (middle row network) included the ISG15-related sub-network, as shown in concentric circle diagrams (purple letters and purple lines). Detailed data are also shown in Supplementary Figs. 6 and 7.

cells of SCA1 patients, ISG15 and Atxn1 staining partially overlapped (Fig. 8c), similar to *Atxn1*-KI mice (Fig. 8e).

Moreover, ISG15 levels were evaluated in peripheral blood samples collected from nine symptomatic SCA1 patients and seven normal controls at the same hospital (Supplementary Data 6). We modified a commercial ELISA kit for ISG15, as it was not suitable for ISG15 in human blood samples, by changing the secondary antibody and revealed an increase of ISG15 level in peripheral blood (Fig. 8d). The data suggested that plasma ISG15 levels were higher in SCA1 patients than in normal controls at the initial sampling time point and at the time point when the plasma ISG15 level was highest among multiple samplings from a SCA1 patient (Fig. 7d). Furthermore, multiple time point samplings from a SCA1 patient revealed

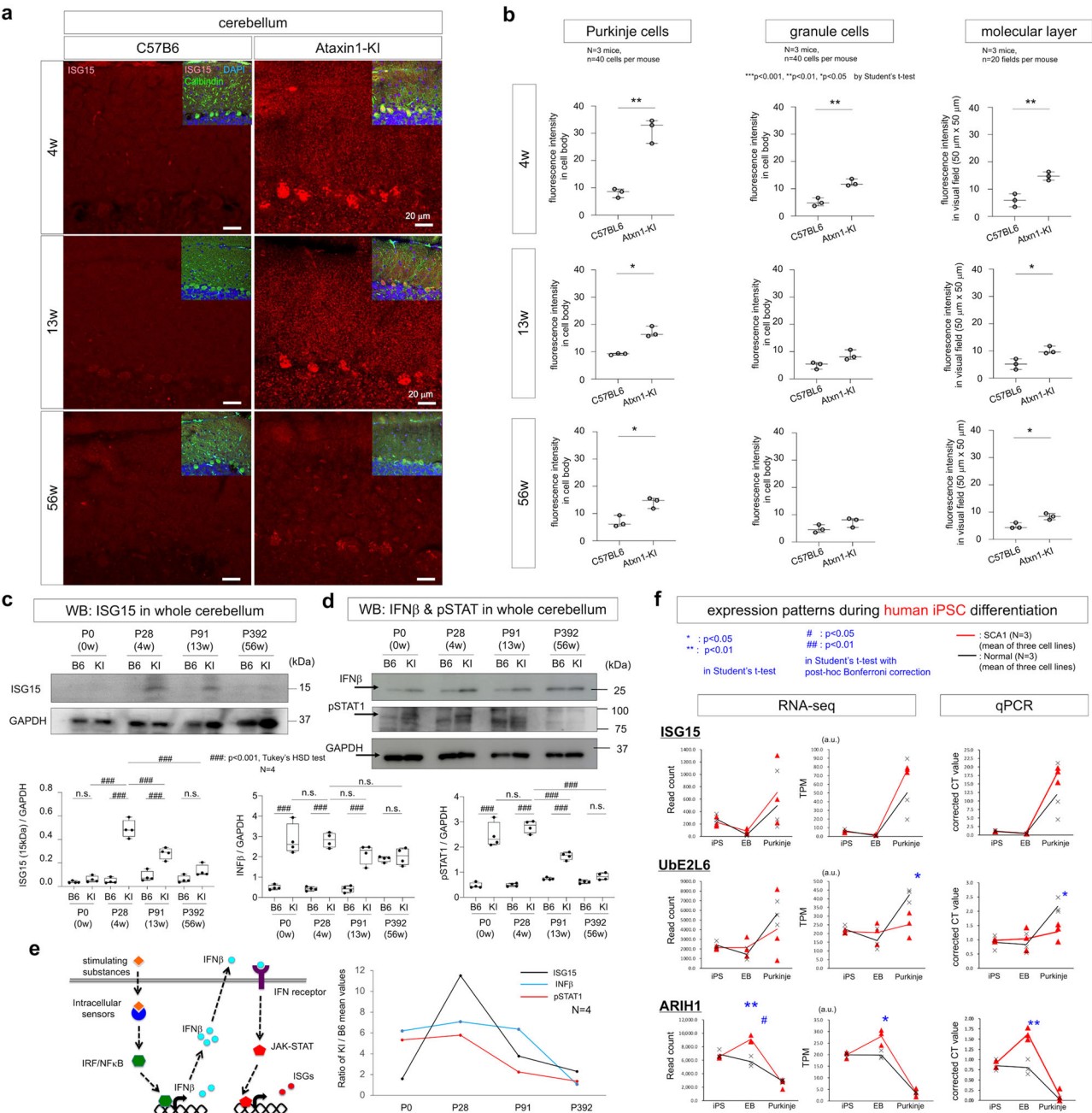

**Fig. 6 | Abnormal increase of ISGylated proteins in SCA1 Purkinje cells. a** ISG15 immunohistochemistry of Atxn1-KI and non-transgenic sibling mouse cerebellar cortexes during aging. Purkinje cells and neuropils of the molecular layer were stained with ISG15 from 4 to 56 weeks of age, while the increase of ISGylated proteins was highest at 4 weeks. **b** Quantitative analyses of ISG15 signals in three layers of the cerebellum. Twenty visual fields per mouse were observed in three C57BL6 and three Atxn1-KI mice. The two-sided Student's *t*-test was used for statistical comparisons. The box plot shows the median and 25–75th percentile, and the whiskers represent data outside the 25–75th percentile range. **c** Western blot analysis of whole cerebellum tissue with anti-ISG15 antibody confirmed increased ISGylated proteins in Atxn1-KI mice. Anti-GAPDH served as loading control. Band intensities were obtained from four independent experiments. Tukey's HSD test was used for multiple comparisons. The box plot shows the median and 25–75th percentile, and whiskers represent data outside the 25–75th percentile range. **d** Western blot analysis of whole cerebellar tissue with anti-IFNβ and anti-pTyr701-STAT1 antibodies. Anti-GAPDH served as loading control. Band intensities were obtained from four independent experiments. Tukey's HSD test was used for multiple

comparisons. The box plot shows the median and 25–75th percentile, and whiskers represent data outside the 25–75th percentile range. **e** The left scheme shows sequential activation of IFNβ, JAK-STAT, and ISGs. The right graph shows a summary of chronological changes of IFNβ, pSTAT, and ISG15 protein expression based on the ratios of their mean values between Atxn1-KI and non-transgenic sibling mice (*N* = 4), which was consistent with the expected scheme. **f** mRNA expression patterns of ISGylation-related genes during differentiation of human iPSC lines (*N* = 3), suggesting the significance of UbE2L6 in abnormal ISGylation associated with SCA1 pathology. The two-tailed Student's *t*-test with the post-hoc Bonferroni correction was used for multiple comparisons. In RNA-seq, *P* = 0.036 in the right panel of UbE2L6 at the Purkinje cell stage, *P* = 0.005 (\*\*) and *P* = 0.014 (#) in the left panel of ARIH1 at the EB stage, and *P* = 0.036 in the right panel of ARIH1 at the EB stage. In qPCR, *P* = 0.021 in the panel of UbE2L6 at the Purkinje cell stage, and P = 0.005 in the panel of ARIH1 at the EB stage. Corrected values mean cycle threshold (CT) values in qPCR that were normalized to GAPDH and referenced to iPSC stage in the calculations. Three independent RNA samples were used for analysis.

**Fig. 7 | ISGylation delays proteosomal protein degradation in SCA1. a** Western blot of whole cerebellum reveals increase of ISGylated proteins in mutant Atxn1-KI mice, which changed during aging (left panel). Red and blue arrows indicate 40 kDa and 100 kDa bands that were changed significantly in mutant Atxn1-KI mice. The same filter was reprobed with anti-Atxn1 and anti-Ub antibodies (right panels). Blue and red arrows indicate 100 and 40 kDa bands, respectively. Black arrows indicate the ISG15 monomer, whose signal remained after washing for reprobing. Anti-GAPDH served as loading control. **b** Reprobing of the same filter with anti-Atxn1 and anti-Ub antibodies. The 100 kDa (blue arrow) and 40 kDa (red arrow) bands were reactive to both antibodies. Small and large black arrowheads indicate residual normal and mutant Atxn1 proteins before degradation. Larger magnified images (right panels) show the bands of normal and mutant Atxn1. **c** Possible patterns of Atxn1 protein modification by both ISG15 and Ub. **d** Western blotting with anti-Ub and anti-Atxn1 antibodies revealed that ISGylated proteins immunoprecipitated from whole cerebellum sample of mutant Atxn1-KI mice included multiple forms of ubiquitinated Atxn1. Thin blue arrows indicate monomeric Atxn1 that is both ISGylated and ubiquitinated. Thick blue arrows and blue arrowheads indicate conjugated Atxn1 via cross-linking by transglutaminases. These high molecular cross-linked Atxn1 species were both ISGylated and ubiquitinated. **e** Super-resolution microscopy of Purkinje cells in Atxn1-KI and normal sibling mice at 4 weeks of age. **f** The protocol for the pulse-chase assay to examine the effect of ISGylation on Ub-proteasome-dependent and autophagy-dependent protein degradation. The lower panel shows the result of the pulse-chase assay. **g** The pulse-chase assay shows protein degradation after protein synthesis is blocked by cycloheximide. Degradation of Myc-Atxn1-86Q was promoted by siRNA-ISG15, while that of FLAG-Ku70 or Myc-Atxn1-33Q was not.

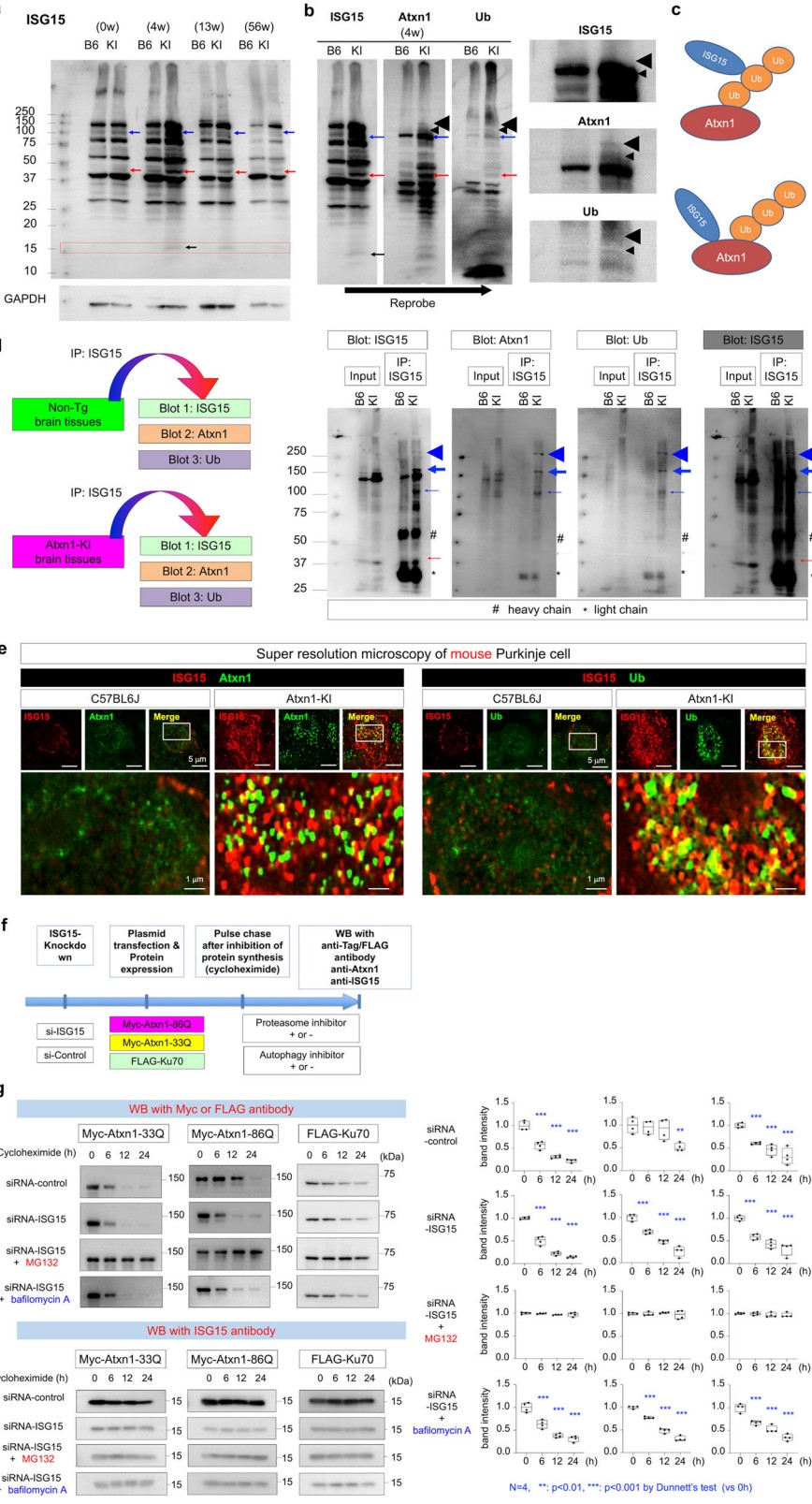

that ISG15 levels declined over time after the first consultation (Fig. 8e, Supplementary Data 6). Relationships between CAG repeat numbers and plasma ISG15 levels were implicated, but statistical significance was not confirmed (Fig. 8f). This might be because the terminal stage pathologies of human patients with 40–60 CAG repeats do not differ. Although disease

severity (Scale for Assessment and Rating of Ataxia (SARA)) and change of disease severity (changes of SARA) seemed to be relevant to plasma ISG15 levels (Supplementary Data 6), further examination with a larger number of samples is necessary to determine the utility of ISG15 as a biomarker of SCA1.

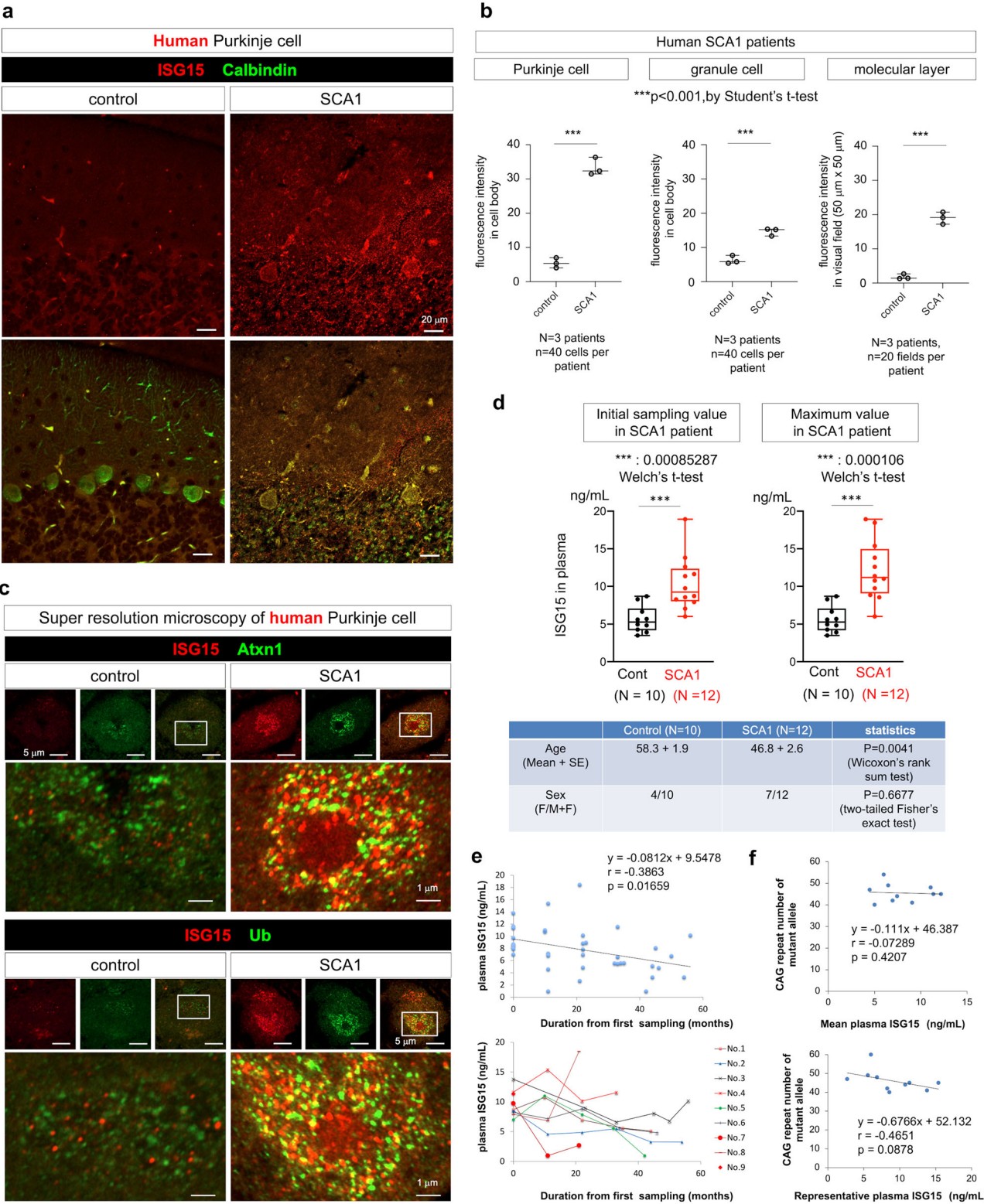

## ISGylation impairs degradation of Atxn1 protein in human Purkinje cells

To confirm the findings obtained in mice, we examined enhanced ISGylation and inhibition of proteasomal degradation of human mutant Atxn1 in cerebellar tissues of a post-mortem patient (28/53 CAG repeats) (Fig. 9). We performed immunoprecipitation with an anti-ISG15 antibody and confirmed that ISGylation of mutant Atxn1 was enhanced (Fig. 9a, b). Ubiquitination of immunoprecipitated ISGylated mutant Atxn1 protein

was also enhanced (Fig. 9a, b). Moreover, we performed cycloheximide chase analysis of human iPSC-derived Purkinje cells. Degradation of mutant Atxn1 was delayed in comparison with that of normal Atxn1, but was enhanced by suppression of ISGylation by siRNA (Fig. 9c, d). Enhanced degradation of mutant Atxn1 by siRNA-ISG15 was blocked by the proteasome inhibitor MG132, but was unaffected by the autophagy inhibitor bafilomycin A (Fig. 9c, d), indicating that ISG15 inhibits Ub-proteasome-dependent degradation of mutant Atxn1 protein in human Purkinje cells.

**Fig. 8 | ISG15 in human SCA1 brains. a** Co-immunostaining of ISG15 and Calbindin. ISG15 signals were increased in the cytosol, neurites, and nucleus of Calbindin[+] Purkinje cells of SCA1 patients in comparison with the control. Representative images of a SCA1 patient with 50 CAG repeats are shown. **b** Quantitative analysis of ISG15 signal intensities in Purkinje cells, granule cells, and molecular layers. Forty cells or twenty visual fields per person were observed in three controls and three SCA1 patients. The two-sided Student's *t*-test was used for statistical comparisons. The box plot shows the median and 25–75th percentile, and whiskers represent data outside the 25–75th percentile range. **c** Super-resolution microscopy of human Purkinje cells in autopsy samples from SCA1 patients or non-neurological disease controls revealed colocalization of ISG15 and Atxn1 and of ISG15 and Ub. **d** An ELISA was developed to measure ISG15 or ISGylated proteins

in plasma of human SCA1 patients and non-neurological disease controls. The two-sided Welch's *t*-test was used for statistical comparison. Blood sampling was performed multiple times in eight of nine SCA1 patients, but only once in normal controls. The maximum value of ISG15 in each SCA1 patient is used in the left graph, and the value of ISG15 at the initial blood sampling is used in the right graph. **e** Relationships between the plasma ISG15 level and period from the first consultation or disease severity (SARA) and between the change of the plasma ISG15 level and the change of disease severity (SARA score) were examined. **f** Relationships between the CAG repeat number and the plasma ISG15 level were examined. Mean values of ISG15 in each patient are used in the upper graph, while the highest values in each patient are used in the lower graph.

## Discussion

In this study, we sought to elucidate the developmental pathology of SCA1 using SCA1 patient-derived iPSCs (Fig. 1, Supplementary Fig. 1) and employing multiple molecular network analyses including iMAD, an originally developed dynamic molecular network analysis of sequentially acquired RNA-seq data during differentiation of SCA1-iPSCs from iPSCs to Purkinje cells. The dynamic molecular network analysis developed in this study, iMAD, revealed a previously unknown pathology of SCA1, i.e., ten core genes at iPSC stage, immune response signaling, and ISG15. Ten core genes included GABRB2, which is closely related to functional development of the cerebellum[44] and TYROBP, which regulates NK cell and microglia activations[45]. TYROBP has been implicated also in AD[46], PD[47], ALS[48], SCA6[49] via interaction with TREM2 in microglia, modulation of NK cells and so on[50–52]. In addition, the core genes included fundamental genes like HIST1H2BF, HIST1H2BH, HIST4H4, ALB, GSTM1, PRSS1, DHRS2 and CCNA1, some of which triggered cytokine-relevant neuroinflammation pathways. These findings suggested existence of ultra-early-phase pathologies in SCA1. In addition, as one of such ultra-early-phase cytokine-relevant pathways, high levels of ISGylated proteins were detected in Purkinje cells of SCA1 model mice. Consistently, the similar increase of ISGylated proteins was confirmed in human SCA1 patients, and ISG15 was upregulated in peripheral blood of human SCA1 patients. These findings suggest that ISG15 is a potential player in SCA1 pathology and a candidate biomarker of human SCA1, although further extensive analyses are necessary to confirm this.

ISG15 was originally identified as a 15 kDa protein increased in Ehrlich ascites tumor cells by IFN stimulation[23]. ISG15 has three major functions[53]. Mature ISG15 generated by cleavage of the C-terminus possesses an LRLRGG Ub-like motif, which is conjugated to the Lysine residue of target proteins in ISGylation. Like ubiquitination, the E1 enzyme (UBE1L), E2 enzyme (UBE2L6/UbcH8/UbcM8), and E3 Ub/ISG15 ligase HERC5, TRIM25 or ARIH1 conduct ISGylation[53]. Intracellular ISG15 exists as a free molecule when it is not used for ISGylation or cleaved from conjugates by USP18-dependent deISGylation[54], and free intracellular ISG15 suppresses IFN1-induced signaling as a feedback mechanism[55]. Our results reveal that both ISGylated proteins and free ISG15 are upregulated in SCA1 pathology in vivo and that enhanced ISGylation affects degradation of mutant Atxn1 protein in Purkinje cells.

In terms of human diseases, ISG15 and ISGylation are implicated in cancer, neurodegeneration, infection, and inflammatory diseases[56], Interestingly, in neurodegenerative disease, ISG15 (also known as Ub cross-reactive protein, UCRP) is upregulated in human A-T fibroblasts (1588: Coriell Cell Repositories, Camden, NJ)[57]. In relevance to the A-T pathology, a research group suggested a negative relationship between proteasome-mediated protein degradation and ISG15 upregulation[58], and ISG15 might also impair mitochondrial quality control[59] and autophagy[60] in A-T cells. Moreover, ISGylation is implicated in Parkinson's disease[61] and amyotrophic lateral sclerosis[62]. Though ISGylation was suggested to enhance E3 Ub ligase activity[61,63] and to promote lysosomal degradation[64].it remains largely unknown how it contributes to their pathologies.

In relevance to neuroinflammation, a research group performed single-cell RNA-seq analysis of MCI and AD cerebral cortex samples collected at autopsy and found nine groups of microglia in terms of gene expression patterns, one of which had high expression levels of ISG15[65]. Another group reported that *ISG15* gene expression and ISGylated proteins are upregulated in demyelinating neurons in experimental autoimmune encephalomyelitis and upon cuprizone-induced demyelination[66]. ISG15 is also released from cells[67], and extracellular ISG15 functions as a cytokine affecting NK and T-cells[68,69] by binding the cell surface receptor LFA-1[70]. Given that involvement of NK and T-cells in neurodegenerative diseases was recently proposed[48,71], the pathological roles of ISG15 mediated by NK and T-cells in SCA1 and other neurodegenerative diseases should be investigated in the future. On the other hand, a possible preventive approach to neurodegenerative diseases by suppression of ISG15 needs attentions because ISG15-deficient patients are susceptible to mycobacterial infection[72].

Homozygous knockout of the USP18 gene results in accumulation of ISG15-conjugated proteins in ependymal cells and increased free ISG15, together with a robust ventricular enlargement cortical atrophy phenotype[73]. However, the study focused on changes to ependymal cells and did not analyze cerebral neurons from the perspective of molecular neuropathology[73]. Therefore, the combined effects of increased levels of intracellular and extracellular ISG15 on neurons in vivo remained unknown until now. Our study addressed this and presented a model in which increased ISGylation inhibited Ub-proteasome-mediated degradation of mutant disease proteins.

Acceptor sites of Ub for ISGylation were intensively analyzed by a previous study, and Lys29 was identified as the site for ISG15 conjugation[38]. Lys29-linked poly-Ub chain is a target of the HECT E3 ligase KIAA10/UBE3C and 26 S proteasome[74–77], indicating that Lys29-linked poly-Ub is a tag for protein degradation. Therefore, ISG15 conjugation to Ub at Lys29 could compete for Lys29-linked poly-Ub chain generation and thus could inhibit protein degradation[38]. Moreover, the Lys29-linked poly-Ub chain of Jmjd2d regulates transcription of interleukin genes in the downstream of Toll-like receptor (TLR) signaling[78–80]. Considering the functional roles of ISG15 conjugation to Ub at Lys29, the increase of ISG15 signals in Purkinje cells of SCA1 model mice and human patients is likely to be a critical event in developmental SCA1 pathology that precipitates events of advanced disease such as protein accumulation/aggregation and brain inflammation.

Several candidate drugs including our AAV-HMGB1[33].are under development, whereas an issue remains how to evaluate the therapeutic effects on SCA1 whose annual progression of clinical symptoms using the SARA is slow[81,82], like the other types of SCA[83–85]. ISG15 was suggested to be a biomarker of breast cancer[86]. However, this is the first study to reveal that ISG15 is a candidate blood biomarker that might be able to evaluate the therapeutic effects of a candidate drug in clinical trials.

In summary, our study revealed a new approach for unveiling the earliest pathologies of neurodegenerative diseases and discovered unexpected molecules that function from the earliest stages of the pathology of SCA1. These findings may spearhead similar studies about the earliest pathologies of neurodegenerative diseases, leading to prevention of neurodegeneration in the future.

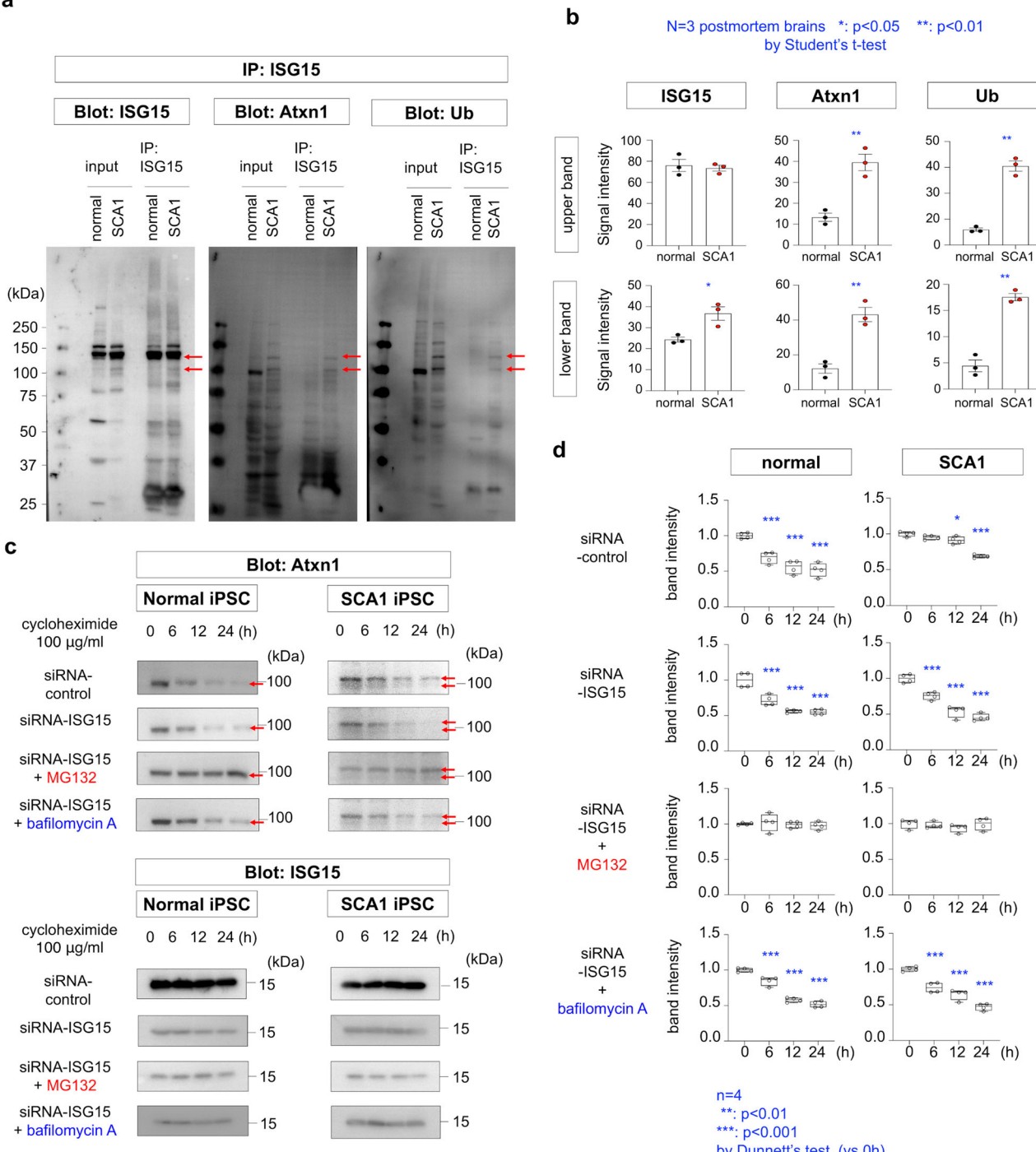

**Fig. 9 | Impaired degradation of ISGylated Atxn1 protein in human Purkinje cells. a** ISGylated proteins were immunoprecipitated with an anti-ISG15 antibody from post-mortem cerebellar tissues of normal control and SCA1 patients and blotted with an anti-ISG15, anti-Atxn1, or anti-Ub antibody. The images were the result with a representative patient (28/53 CAG repeats). **b** Quantitative analysis of the signal intensities of Atxn1 or ubiquitinated Atxn1 bands in immunoprecipitates obtained using an anti-ISG15 antibody. $N = 3$ (see Methods). **c** Chase assay of Atxn1 protein degradation after inhibition of protein translation by cycloheximide in human iPSC-derived Purkinje cells. Degradation of mutant Atxn1 was delayed in comparison with that of normal Atxn1 and was normalized by siRNA-ISG15. The enhancement of protein degradation by siRNA-ISG15 was blocked by the proteasome inhibitor MG132, but was unaffected by the autophagy inhibitor bafilomycin A. **d** Quantitative analysis of the signal intensities of Atxn1 bands during the chase assay. $N = 4$.

## Methods

### iPSC culture and differentiation to pan-neurons

SCA1-iPSCs and normal iPSCs were differentiated to pan-neurons[87]. SCA1-iPSCs and normal iPSCs were cultured in TeSR-E8 medium (STEMCELL Technologies, BC, Canada) with 10 μM Y27632 (253-00513, Wako, Osaka, Japan). After 24 h, medium was changed to Stem Fit (AK02N, Ajinomoto, Tokyo, Japan) containing 5 μM SB431542 (13031, Cayman Chemical, Ann Arbor, MI, USA), 5 μM CHIR99021(13122, Cayman Chemical, Ann Arbor, MI, USA), and 5 μM dorsomorphin (044-33751, Wako, Osaka, Japan). After 5 days, iPSCs were dissociated with TrypLE Select (12563-011, Thermo Fisher Scientific, MA, USA). Neurospheres were then cultured in KBM medium (16050100, KHOJIN BIO, Saitama, Japan) with 20 ng/mL

Human-FGF-basic (100-18B, Peprotech, London, UK), 10 ng/mL Recombinant Human LIF (NU0013-1, Nacalai, Kyoto, Japan), 10 μM Y27632 (253-00513, Wako, Osaka, Japan), 3 μM CHIR99021 (13122, Cayman Chemical, Ann Arbor, MI, USA), and 2 μM SB431542 (13031, Cayman Chemical, Ann Arbor, MI, USA) for 10 days. Finally, neurospheres were dissociated and seeded onto chambers coated with poly-L-ornithine (P3655, Sigma-Aldrich, St. Louis, MO, USA) and laminin (23016015, Thermo Fisher Scientific, Waltham, MA, USA), and cultured in DMEM/F12 (D6421, Sigma-Aldrich, St. Louis, MO, USA) supplemented with B27 (17504044, Thermo Fisher Scientific, Waltham, MA, USA), Glutamax (35050061, Thermo Fisher Scientific, Waltham, MA, USA), and penicillin/streptomycin (15140-122, Thermo Fisher Scientific, Waltham, MA, USA) for 14 days.

### Differentiation of iPSCs to Purkinje cells

SCA1-iPSCs and normal iPSCs were differentiated to Purkinje cells[88]. To form EBs, iPSCs were dissociated with TrypLE Select (12563-011, Thermo Fisher Scientific, MA, USA), and 24,000 cells were aggregated by centrifugation at $200 \times g$ for 2 min in 96-well U-bottomed culture plates (650-180, Greiner, Kremsmünster, Austria) coated with Lipidure (CM5206, Nichiyu, Tokyo, Japan). Cells were cultured with gfCDM/insulin medium, 1:1 Iscove's modified Dulbecco's medium (12440053, Thermo Fisher Scientific, Waltham, MA, USA), and Ham's F-12 (11765054, Thermo Fisher Scientific, Waltham, MA, USA) with 7 μg/mL insulin (I5500, Sigma-Aldrich, St. Louis, MO, USA); 1x chemically defined lipid concentrate (11905031, Thermo Fisher Scientific, Waltham, MA, USA); 15 μg/ml apo-transferrin (T4382, Sigma-Aldrich, St. Louis, MO, USA); 450 μM mono-thioglycerol (195-15791, Thermo Fisher Scientific, Waltham, MA, USA); 5 mg/mL BSA (A7608, Sigma-Aldrich, St. Louis, MO, USA), Glutamax (35050061, Thermo Fisher Scientific, Waltham, MA, USA), and penicillin/streptomycin (15140-122, Thermo Fisher Scientific, Waltham, MA, USA); 20 μM Y-27632 (253-00513, Wako, Osaka, Japan); and 10 μM SB431542 (13031, Cayman Chemical, Ann Arbor, MI, USA). After 2 days, 50 ng/mL recombinant human FGF2 (233-FB-025, R&D systems, MN, USA) was added to culture medium. After 21 days, EBs were transferred to 10 cm Petri dishes (1020-100, Iwaki, Shizuoka, Japan) coated with Lipidure (CM5206, Nichiyu, Tokyo, Japan) and cultured for 14 days in Neurobasal/N2 medium, Neurobasal (21103-049, Thermo Fisher Scientific, Waltham, MA, USA) with N2 supplement (17502048, Thermo Fisher Scientific, Waltham, MA, USA), Glutamax (35050061, Thermo Fisher Scientific, Waltham, MA, USA), and penicillin/streptomycin (15140-122, Thermo Fisher Scientific, Waltham, MA, USA).

EBs were dissociated and cocultured with rhombic lip (RL) cells isolated from cerebellums of E14 Slc:ICR mice to induce differentiation into Purkinje cells. Briefly, RLs and EBs were dissociated with TrypLE Select (12563-011, Thermo Fisher Scientific, MA, USA) and cocultured in DMEM/F12 medium (11330032, Sigma-Aldrich, St. Louis, MO, USA) with 10% FBS, N2 supplement (17502048, Thermo Fisher Scientific, Waltham, MA, USA), and penicillin/streptomycin (15140-122, Thermo Fisher Scientific, Waltham, MA, USA). A total of $1.0 \times 10^6$ cells at cell ratio = 1:10 (EB: RL) with 80 μL medium were seeded on chambers coated with poly-L-lysine (P1524-25MG, Sigma-Aldrich, St. Louis, MO, USA) and laminin (23016015, Thermo Fisher Scientific, Waltham, MA, USA). After incubation for 6 h, DMEM/F-12 supplemented with N2 (17502048, Thermo Fisher Scientific, Waltham, MA, USA), 100 μg/mL BSA (A7608, Sigma-Aldrich, St. Louis, MO, USA), 50 ng/mL human BDNF (248-BDB-010/CF, R&D systems, MN, USA), 50 ng/mL human NT3 (267-N3-005/CF, R&D systems, MN, USA), and penicillin/streptomycin (15140-122, Thermo Fisher Scientific, Waltham, MA, USA) medium was added and cultured for 10 days.

EB-derived differentiating cells and RL-derived cells were cultured in cell culture insert dishes (140640, Thermo Fisher, Waltham, MA, USA) in which the two types of cells could be separated to avoid RL contamination of RNA-seq samples. The cell culture insert dish was coated with poly-L-lysine (P1524-25MG, Sigma-Aldrich, St. Louis, MO, USA) and laminin

(23016015, Thermo Fisher Scientific, Waltham, MA, USA), embryonic body-derived cells were then seeded on the lower well, and RL-derived cells were seeded on a polycarbonate insert with a 4 μm pore. The culture medium was the same as described above.

### Analysis of differentiated iPSCs

To quantify the cell growth rate of SCA1-iPSCs and normal iPSCs, 30,000 iPSCs were seeded per well on Day 0. After 2, 4, 6, or 8 days, cells were collected, dissociated by 0.5 x TrypLE Select (12563-011, Thermo Fisher Scientific, MA, USA) and counted by using Burker-Turk hemocytometer. To quantify the size of iPSC-derived EBs, images of EBs were taken by microscope (IX70, Olympus) on day 1, 7, 14 and 21, and the 2D areas of EBs reflecting their 3D sizes were measured by ImageJ software (version 1.50, NIH, MD, USA).

AAV-HMGB1-EGFP or AAV-EGFP were infected into differentiated pan-neurons (MOI 2000). Twelve days after AAV infection, cells were fixed with 1% paraformaldehyde in PBS for 30 min. After blocking in PBS containing 10% FBS for 30 min, cells were stained with the following primary and secondary antibodies: mouse anti-βIII-tubulin 1 (1:2000 for 16 h at 4 °C, #T8660 Sigma-Aldrich, St. Louis, MO, USA), rabbit anti-PSD95 (1:1000 for 16 h at 4 °C, 3409, Cell Signaling Technology, Danvers, MA, USA), donkey anti-mouse IgG Alexa 488-conjugated (1:600 for 1 h at room temperature, #715-545-150, Jackson ImmunoResearch Laboratories, West Grove, PA, USA), and donkey anti-rabbit IgG Alexa488-conjugated (1:1000 for 1 h at room temperature, A-21206, Thermo Fisher Scientific). The dendritic spine was assessed after acquisition of image by confocal microscopy (FV1200 laser scanning microscope, Olympus, Tokyo, Japan). We used x40 objective lens (UPLSAPO40X2 (NA:0.95)). The dendritic spine density was measured by ImageJ software (version 1.50, NIH, MD, USA). After calibration of scale information, dendrite (Tuj-1-immunostained image) were manually traced and calculate the dendrite length in observing image window. Next, the spine (PSD95-immunostained dots) was counted, and the dendritic spine density was defined as the number of spines in 1 μm length of a dendrite, calculated by dividing a spine number in one dendrite by the length of it.

Differentiated Purkinje cells were fixed with 1% paraformaldehyde in PBS for 30 min. Cells were incubated with 10% FBS followed by incubation with primary and secondary antibodies as follows: mouse anti-Calbindin (1:2000 for 16 h at 4 °C, C9848, Sigma-Aldrich, St. Louis, MO, USA) and donkey anti-mouse IgG Alexa 488-conjugated (1:1000 for 1 h at room temperature, A-21202, Thermo Fisher Scientific).

### Immunohistochemistry

Mice brain tissues at different timepoints (P0, P28, P91, and P392) were fixed with 4% paraformaldehyde in 0.1 M phosphate buffer for 12 h and embedded in paraffin. Sagittal sections were deparaffinized in xylene and rehydrated in ethanol. For antigen retrieval, sections were incubated in Tris-EDTA solution pH 9.0 (100 mM Tris-base and 10 mM EDTA) at 121 °C for 15 min. Human brain paraffin sections of SCA1 patients or control sections were also processed. Sections were then incubated with 0.5% triton-X 100 in PBS for 20 min to perform permeabilization. Next, sections were incubated with 10% FBS in PBS for 30 min and were incubated with primary and secondary antibodies as follows: rabbit anti-ISG15 (1:100 for 16 h at 4 °C, HPA004627, Sigma-Aldrich, St. Louis, MO, USA), mouse anti-Calbindin (1:2000 for 16 h at 4 °C, C9848, Sigma-Aldrich, St. Louis, MO, USA), mouse anti-Atxn1 (1:100 for 16 h at 4 °C, MABN37, Millipore, Burlington, MA, USA), mouse anti-Ub (1:100 for 16 h at 4 °C, #3936, Cell Signaling technology, Danvers, MA, USA), donkey anti-mouse IgG Alexa 488-conjugated (1:600 for 1 h, #715-545-150, Jackson ImmunoResearch Laboratories, West Grove, PA, USA), and donkey anti-rabbit IgG Cy3-conjugated (1:600 for 1 h, #711-165-152, Jackson ImmunoResearch Laboratories, West Grove, PA, USA). Nuclei were stained with DAPI (0.2 μg/mL in PBS, D523, DOJINDO Laboratories, Kumamoto, Japan). Z-stacked images (0.5 μm interval x 5 slices) were acquired from cerebellar cortex (Lobule IV/V) using a confocal microscope (FV1200IXGP44, Olympus, Tokyo, Japan) and a

super-resolution microscope (LSM980 with Airyscan 2, Zeiss, Oberkochen, Germany). Signal intensities were measured using ImageJ software.

## Western blot analysis

Male mice brain tissues at different time points (P0, P28, P91, and P392) were homogenized using the BioMasher II (#893062, Nippi, Tokyo, Japan) with RIPA buffer (10 mM Tris-HCl pH 7.5, 150 mM NaCl, 1 mM EDTA, 1% Triton-X 100, 0.1% SDS, 0.1% DOC, and 1:250 volume Protease Inhibitor Cocktail (#539134, Calbiochem, San Diego, CA, USA)). Homogenates were centrifuged at 12,000 $g$ for 10 min, and supernatants were added to equal volumes of sample buffer (0.1 M Tris-HCl pH 6.8, 4% SDS, 20% glycerol, 0.05% BPB, and 12% β-mercaptoethanol) and boiled at 100 °C for 10 min. Samples were subjected to SDS-PAGE and transferred onto PVDF membrane. After blocking the membranes with 5% skim milk in TBST (20 mM Tris-HCl pH 7.5, 150 mM NaCl, 0.05% Tween-20) for 1 h, membranes were incubated with primary and secondary antibodies as follows: rabbit anti-ISG15 (1:1000 for 3 h at room temperature, HPA004627, Sigma-Aldrich, St. Louis, MO, USA), mouse anti-GAPDH (1:3000 for 16 h at 4 °C, MAB374, Merck, Darmstadt, Germany), mouse anti-Atxn1 (1:1000 for 3 h at room temperature, MABN37, Millipore, Burlington, MA, USA), mouse anti-Ub (1:1000 for 16 h at 4 °C, #3936, Cell Signaling Technology, Danvers, MA, USA), mouse anti-Myc (1:3000 for 1 h at room temperature, M047-3, MBL, Aichi, Japan), rabbit anti-FLAG (1:3000 for 1 h at room temperature, F7425, Sigma, St. Louis, MO, USA), sheep anti-mouse IgG HRP conjugated (1:3000 for 1 h, NA931, Cytiva, Tokyo, Japan), and rabbit anti-IgG HRP conjugated (1:3000 for 1 h, NA934, Cytiva, Tokyo, Japan). Proteins were detected using Amersham ECL select (RPN2235, Cytiva, Tokyo, Japan) on an Image-Quant luminescence image analyzer LAS500 (Cytiva, Tokyo, Japan). Signal intensities were measured using ImageJ software.

## RNA sequencing

SCA1-iPSCs and normal iPSCs were collected and homogenized in 350 μL RNA RLT buffer (Qiagen)/0.01% 2-mercaptoethanol (Wako, Tokyo, Japan). Total RNA was purified with RNeasy mini kit (Qiagen). To eliminate genomic DNA contamination, on-column DNA digestion was conducted for each sample with DNase I (Qiagen). Prepared RNA samples were subjected to a HiSeq-based RNA-seq by TAKARA (700 million bp reads).

## Analysis of differential gene expression

Gene expression profiles of each sample were evaluated by the number of short reads that were mapped onto gene coding sequences in the reference human genome assembly hg38. Differential expression genes were analyzed with DESeq2[34]. Log₂FC (Fold Change) between SCA1-derived and normal cells was calculated by DESeq2, and the difference of gene expression was determined at $|\mathrm{Log_2FC}| > 0.5$.

## Construction of the pathological network

To generate the pathological network based on PPI, UniProt accession numbers were added to genes identified in RNA-seq-based gene expression analysis. The pathological PPI network was constructed by connecting genes using the integrated database of the Genome Network Project (GNP) (https://cell-innovation.nig.ac.jp/GNP/index_e.html), which includes BIND, BioGrid (http://www.thebiogrid.org/), HPRD, IntAct (http://www.ebi.ac.uk/intact/site/index.jsf), and MINT. A database of GNP-collected information was created on the Supercomputer System available at the Human Genome Center of the University of Tokyo.

## Static molecular network analysis

To create a static molecular network, statistically significantly changed molecules were connected at two neighboring time points based on interactions in the PPI database (an integrated database collected by GNP, which includes BIND, BioGrid (http://www.thebiogrid.org/), HPRD, IntAct (http://www.ebi.ac.uk/intact/site/index.jsf), and MINT), without considering their cause-result relationships. Each network starting from a changed molecule was expanded step-by-step from one-hop (directly linked) to six

extra connections, and the degree of significance at each expansion step was evaluated by calculating the z-score of their ratio of changed nodes.

## Dynamic molecular network analysis

To gain insights into the dynamics of the pathological molecular network, a PPI-based chronological molecular network was constructed by connecting two proteins between the two neighboring time points using the integrated PPI database.

To estimate the impact of a significantly differentially expressed gene to the future time point, a gene in a certain time point was connected to a set of genes in the next time point based on the PPI database (an integrated database collected by GNP, which includes BIND, BioGrid (http://www.thebiogrid.org/), HPRD, IntAct (http://www.ebi.ac.uk/intact/site/index.jsf), and MINT) and defined as downstream genes. The magnitude of impact of a gene at the first time point (defined as "r") was calculated by the ratio of downstream genes that were significantly changed in mRNA expression levels to all genes at the second time point. The selection of genes was performed based on comparison of impact of a specific gene (specific impact) and total impact at the second time point. The statistical significance of the impact was examined using two-tailed Fisher's exact test with post-hoc Benjamini-Hochberg procedure (adjusted $p$-value < 0.05, red dots). The statistical significance of mRNA expression change was examined using log₂FC between SCA1 and normal cells ($|\mathrm{Log_2FC}| > 0.5$, blue dots). A digraph was created from a significantly differentially expressed gene at the original time point (iPSCs) to significant genes at the end point (Purkinje cells) via significant genes at the intermediate time point based on the impact analysis. The digraph predicts the original gene whose change at the initial time point leads to molecular changes at the final time point.

## GO enrichment analysis

To select cytokine-relevant genes, Gene Ontology (GO) enrichment analysis was performed using clusterProfiler[89] package in R. A list of genes included in a selected pathway was used as input of enrichGO function of cluster-Profiler. From all the enrichment results, GO terms related to cytokine were selected to extract a list of cytokine-relevant genes. The GO terms related to cytokine were searched by keyword "cytokine", and thereafter terms that were not related to cytokine such as "cytokinesis" were excluded.

## Meta-analysis of SCA1 mouse models

A search for transcriptomic studies of SCA1 in *Homo sapiens* and *Mus musculus* in the NCBI Sequence Read Archive (SRA; https://www.ncbi.nlm.nih.gov/sra) was performed on August 7, 2023 using the keywords "spinocerebellar ataxia type 1" and "SCA1". Eight studies (PRJNA305316, PRJNA422988, PRJNA472147, PRJNA472754, PRJNA503578, PRJNA688073, PRJNA871289, and PRJNA903078) that include RNA-seq data and raw sequence information generated from cerebellar tissues of SCA1 mouse models were found, while no studies with human RNA-seq data were found.

Raw sequence data were downloaded from NCBI SRA using the SRA Toolkit (https://github.com/ncbi/sra-tools) and mapped to the *M. musculus* genome assembly GRCm38/mm10 using HISAT2. Gene expression was calculated using the featureCounts function from Subread v1.5.2. Gene expression differences between the SCA1 and control groups were tested using Welch's $t$-test. Gene expression changes with a $p$-value ≤ 0.05 were considered significant.

## Quantitative RT-PCR

RNA was isolated from iPSCs, EBs and Purkinje cells with TRIzol RNA Isolation Reagents (15596026, Thermo Fisher Scientific, MA, USA). Reverse transcription was performed by using the SuperScript VILO cDNA Synthesis kit (11754-250, Invitrogen, Carlsbad, CA, USA). Quantitative PCR analyses were performed with the 7500 Real-Time PCR System (Applied Biosystems, Foster City, CA, USA) by using THUNDERBIRD SYBR qPCR Mix (QPS-201, TOYOBO, Osaka, Japan) and assessed by the standard curve method. The primer sequences were:

ISG15, forward primer: 5′-CGCAGATCACCCAGAAGATCG-3′ and reverse primer: 5′- TTCGTCGCATTTGTCCACCA-3′

UbE2L6, forward primer: 5′-GTGGCGAAAGAGCTGGAGAG-3′ and reverse primer: 5′ -ACACTGTCTGCTGGTGGAGTTC- 3′

ARIH1, forward primer: 5′-CAGGAGGAGGATTACCGCTAC-3′ and reverse primer: 5′-CTCCCGGATACATTCCACCA-3′

GAPDH, forward primer: 5′-AGATCATCAGCAATGCCTCCTG-3′ and reverse primer: 5′-ATGGCATGGACTGTGGTCATG-3′

PCR conditions for amplification were 40 cycles of 95 °C for 1 min for enzyme activation, 95 °C for 15 sec for denaturation, and 60 °C for 1 min for extension. The expression levels of ISG15, UbE2L6 and ARIH1 were corrected by GAPDH.

## Immunoprecipitation

Frozen mouse brains (male, P28) were lysed with TNE buffer (10 mM Tris-HCL (pH 7.5), 150 mM NaCl, 1 mM EDTA, and 1% NP-40) and collected by centrifugation ($15,000 \times g \times 10$ min). Aliquots (100 μg protein in cerebellar tissue lysate) were then incubated 1 h with a 50% slurry of protein G-sepharose beads. After centrifugation ($2000 \times g \times 3$ min), the supernatants were incubated with 1 μg rabbit anti-ISG15 antibody (aHPA004627, Sigma-Aldrich, St. Louis, MO, USA) overnight at 4 °C. Reactants were then incubated with Protein G-sepharose beads for 4 h, washed with TNE buffer, and eluted by sample buffer. For double-precipitation, samples were incubated with 2 μg biotin-labeled mouse anti-Ub (#3936, Cell Signaling technology, Danvers, MA, USA) overnight at 4 °C, followed by incubation with streptavidin beads (TrueBlot(R) Streptavidin Magnetic Beads, S000-18-5, Rockland, Pottstown, PA, USA). The collected samples were further incubated with 2 μg mouse anti-Atxn1 (MABN37, Millipore, Burlington, MA, USA) overnight at 4 °C. Then, reactants were incubated with Protein G-sepharose beads for 4 h, washed with TNE buffer, and eluted in sample buffer.

## Cycloheximide chase assay

HeLa cells were seeded at a density of $4 \times 10^5$ cells/well in a 6-well plate (3516, Corning, Glendale, AZ, USA) and transfected with 30 pmol human ISG15-siRNA (sc-43869, Santa Cruz Biotechnology, Dallas, TX, USA) or scrambled siRNA (SR30004, OriGene, Rockville, MD, USA) using 4 μL Lipofectamine RNAiMAX (13778-075, Thermo Fisher Scientific, Waltham, MA, USA). At 24 h after siRNA transfection, 2.5 μg myc-Ataxin1-33Q, myc-Ataxin1-86Q, or FLAG-Ku70 plasmid was transfected using 5 μL Lipofectamine 2000 (11668-019, Thermo Fisher Scientific, Waltham, MA, USA). At 48 h after siRNA transfection, 100 nM Bafilomycin A1 (BVT-0252-C100, AdipoGen, Liestal, Basel-Landschaft, Switzerland) or 5 μM MG132 (139-18451 Wako, Osaka, Japan) was added to the culture medium in order to inhibit autophagy or proteasome-dependent protein degradation, respectively. At 49 h, 100 μg/mL cycloheximide (033-20993, Wako, Osaka, Japan) was added to the culture medium in order to inhibit protein synthesis. The cells were collected at 0, 6, 12, and 24 h after addition of cycloheximide, lysed with RIPA buffer (10 mM Tris-HCl pH 7.5, 150 mM NaCl, 1 mM EDTA, 1% Triton-X 100, 0.1% SDS, 0.1% DOC, and 1:250 volume Protease Inhibitor Cocktail (#539134, Calbiochem, San Diego, CA, USA)), and centrifuged at 12,000 g for 10 min. The supernatants were mixed with equal volumes of sample buffer (0.1 M Tris-HCl pH6.8, 4% SDS, 20% glycerol, 0.05% BPB, and 12% β-mercaptoethanol), boiled at 100 °C for 10 min, and subjected to SDS-PAGE.

## Atxn-1 knock-in mice

Mutant *Atxn1*-KI mice (Sca1$^{154Q/2Q}$ mice) were generously gifted by Prof. Huda Y. Zoghbi (Baylor College of Medicine, TX, USA)[39]. The backcrossed strain with C57BL/6 mice were further crossed with C57BL/6 female mice more than 10 times in our laboratory. The number of CAG repeats was checked by fragment analysis using the following primers: forward (5'–CACCAGTGCAGTAGCCTCAG–3', labeled with 6-carboxy-fluorescein) and reverse (5'–AGCTCTGTGGAGAGCTGGAA–3'). Mice were maintained under suitable humidity (around 50%) at 22 °C with a 12 h light-dark cycle. We have complied with all relevant ethical regulations for animal use.

## Human materials

Cerebellar specimens collected at autopsy from three SCA1 patients and three control patients without neurological disorders (lung cancer, leukemia, and cholangiocarcinoma) were used. The details of the SCA1 patients (51-year-old female, 54-year-old female, and 50-year-old male) were described previously[90–92]. Their CAG repeat expansion in the *Atxn1* gene was confirmed by PCR and their numbers of CAG repeats were around 50, although the exact numbers were not determined by fragment analysis or Sanger sequencing. Human plasma samples were acquired from SCA1 patients with a PCR-based genetic diagnosis or control patients without neurological disorders. Essential information about the SCA1 patients and controls is shown in Fig. 8D. Other clinical information is not linked with samples according to ethics regulations. All ethical regulations to human research participants were followed.

## Human ISG15 ELISA

In total, 100 μL of human plasma samples that had been diluted 2-fold with saline (OTSUKA normal saline 20 mL, Otsuka Pharmaceutical Factory, Tokushima, Japan) was added to a 96-well plate precoated with an anti-ISG15 antibody (#CY-8085, CircuLex Human ISG15 ELISA Kit, MBL, Tokyo, Japan) and incubated for 16 h at 4 °C. Plates were washed and subsequently incubated with a peroxidase-conjugated anti-ISG15 antibody (Atlas Antibodies, HPA004627-100UL, Bromma, Sweden) for 2 h at room temperature. For detection, Substrate Reagent (#CY-8085, CircuLex Human ISG15 ELISA Kit, MBL, Tokyo, Japan) was added to each well. The reaction was terminated with Stop Solution (#CY-8085, CircuLex Human ISG15 ELISA Kit, MBL, Tokyo, Japan), and absorbance at 450 nm was measured on a microplate reader (SPARK 10 M, TECAN, Grodig, Austria). A standard curve was generated using 0, 1.5, 3, 6, and 12 ng/mL Recombinant Human ISG15 (UL-601-500, R&D Systems, Minneapolis, MN, USA) diluted with Sample Diluent (326078738, HMGB1 ELISA Kit Exp, Shino-test, Tokyo, Japan).

## Statistics and Reproducibility

We analyzed three iPSC lines derived from two SCA1 patients for RNA-seq-based gene expression analysis. For meta-analysis using SCA1 model mice, we collected RNA-seq data from 3 to 17 mice in each time point from NCBI SRA database. Statistical analyses for biological experiments were performed using Graphpad Prism 8. Biological data following a normal distribution are presented as the mean ± SEM, with Tukey's HSD test or Dunnett's test for multiple group comparisons or with Welch's *t*-test for two group comparisons. The distribution of observed data was depicted with box plots, with the data also plotted as dots. Box plots show the medians, quartiles, and whiskers, which represent data outside the 25th–75th percentile range. To obtain each data, we performed biologically independent experiments. The number of samples was indicated in each figure and figure legends.

## Ethics statement

This study was performed in strict accordance with the recommendations of the Guide for the Care and Use of Laboratory Animals of the Japanese Government and National Institutes of Health. All experiments were approved by the Committees on Gene Recombination Experiments, Human Ethics, and Animal Experiments of the Tokyo Medical and Dental University (G2018-082C3, 2014-5-3, and A2021-211A). Human samples including post-mortem brains were provided with informed consent and their use was approved by the Committees on Human Ethics (O2020-002-03).

## Reporting summary

Further information on research design is available in the Nature Portfolio Reporting Summary linked to this article.

## Data availability

Source data underlying the figures are provided in the Supplementary Data 7 file. Raw RNA-seq data generated in this study have been deposited in the NCBI SRA database (accession code PRJNA1029787). Other data obtained in the current study are available from the corresponding author upon reasonable request.

## Code availability

A custom code for iMAD-based molecular network analysis is available via Zenodo[93] (https://doi.org/10.5281/zenodo.10820714).

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

## Acknowledgements

We thank Prof. Huda Y. Zoghbi (Baylor College of Medicine) for providing *Atxn1*-KI (Sca1$^{154Q/2Q}$) mice; Prof. Yoshinobu Eishi (Tokyo Medical and Dental University), Prof. Masanobu Kitagawa (Tokyo Medical and Dental University), and Dr. Saburo Yagisita (Kanagawa Rehabilitation Center) for providing human control and SCA1 pathological samples; and Dr. Takuya Tamura and Dr. Emiko Yamanishi (Department of Neuropathology, Tokyo Medical and Dental University) for their technical support. This work was supported by grants to H.O., including 21ek0109527s0201, 22ek0109527s0202, 23ek0109527s0203, Brain/MINDS and 23gm1910001h0001 from the Japanese Agency for Medical Research and Development (AMED); a Grant-in-Aid for Scientific Research on Innovative Areas (Foundation of Synapse and Neurocircuit Pathology, 22110001/22110002) from the Ministry of Education, Culture, Sports, Science, and Technology of Japan (MEXT); and a Grant-in-Aid for Scientific Research A (16H02655, 19H01042, 22H00464) from the Japanese Society for the Promotion of Science (JSPS). This work is dedicated to Prof. Katsutaka Nagai and Prof. Ichiro Kanazawa.

## Author contributions

H.H., Y.Y.: Data acquisition, writing—original draft, and draft revision. S.S., Y.H.: Human resource and data collection. I.Y., H.S., Y.S.: Experimental conceptualization, human resource, and data collection. H.K., H.Okano: Experimental conceptualization and iPSC generation. K.F., H.T.: Experimental conceptualization, data acquisition, writing—original draft, and draft revision. H.Okazawa: Project conceptualization, supervision, funding acquisition, project administration, writing—original draft, and draft revision.

## Competing interests
The authors declare no competing interests.
