## [Peer Review File · Communications Biology]

Reviewers' comments:

Reviewer #1 (Remarks to the Author):

Homma et al described a molecular study on Spinocerebellar ataxia type-1 models which includes patient derived cell lines, SCA1-Mice model and validation exercise involving peripheral blood. Overall, a voluminous work that highlights the role ISGylation of ATXN1 as marker and a process of protein modification (ATXN1 here) in SCA1 pathology. There are several caveats in the work about the presentation of various findings at each level of model system involvement. The study has three components mainly, i) generation of purkinje cell lines from lymphocytes derived iPSc, ii) RNAseq of these cell lines and iii)Atxn1 knock-in mice model for the validation. Overall these appears somewhat connected however majorly disunited for

a) iPSC derived pan-neuronal lineage and purkinje lines: major findings of this work is at morphological level that exhibit impaired neurite length and spine density of the pan-neurons and purkinje lines. While at RNAseq the purkinje line was utilized for deriving network based molecular perturbation with connectome initiated at iPSC level. The finding of HMGB1 mediated rescue of spine density was well connected with RNAseq based network analysis.

b) Molecular Dynamic network analysis of RNAseq: Here, the temporal model of analysis was taken from iPSC lineage-Embroid body-Purkinje cells to decipher the maximally disturbed network in SCA1 model and thus extrapolated to ISGylation.

c) ISGylation model was followed in Atxn-1 knock-in mice

Major concerns

a) For patient derived lines component of the study

1. The fundamental point of full penetrance model of SCA1 requires uninterrupted CAG repeats of length 39-44 in human studies (<https://www.ncbi.nlm.nih.gov/books/NBK1184/>). The validity of the model per say seems less precise in terms of the repeat sizes of the PBMC were in range of 41-47 with CAT interruption. The uninterrupted length of CAG in the model is only 28-36 not at least 39-44. Moreover the repeat calculation is wrongly depicted in Fig S1A (Table, peripheral blood vs clones??)

2. The major drawback of the work is correlation of the data with small number of RNAseq data (n=2 patient line Vs n=1 control line) to derive the most significant expressed for the downstream analysis.

The basic architecture of RNAseq data analysis was not shown, though volcano plot and pathway analysis of diff expressed genes have been presented. No detailed list is available to decipher the key dystregulated genes in SCA1 lines vs control. Has data been deposited in SRA (NCBI)? The raw matrix of data QC need to shown in supplementary text/table.

3. Figure S2, line "176-79, normal and SCA1 EBs to pan-neurons, and measured neurite extension length (Supplementary Figure 2A) and density of PSD95-positive spine-like protrusions as indicators of neuronal differentiation (Supplementary Figure 2B)". requires to be corrected.

4. Line 239-242, needs to re-phrased.

b) Molecular dynamic network analysis: This seems an interesting model however in absence of through validity check of the network and its parameter in view of the temporal nature of the limited dataset in the current work analysis the usefulness of the analysis and outcome is limited. There are different sort

of approach possibly require to draw major conclusion on the network analysis because this dataset is majorly a RNAseq data and in isolation will always be noisy. The first (mentioned above) component of study gave a lead around cerebellin and then luciferase reporter assay was designed and that typically shows a regulatory framework. Thus how similar framework be equally applied while the data also an temporal dynamic component. There are several factors which can influence the network, its edges and nodes. More robust analysis using meta-analysis of publically available dataset (of another PolyQ disease) can be brought in support of the validity (proof of principle) of this type of network analysis. The temporally dynamic data are best analyzed using Boolean networks.

c) ISGylation model was followed in Atxn-1 knock-in mice: this component of work is interesting independently and owing to widely believed known roles of ISGylation in various pathologies. Subsequently, to the mice work validation of ISG15 was performed in post-mortem brain samples. Line of clarification required for the observation on level of increased ISG15 level which was observed developmentally in mice, while, on the contrast post-mortem brain tissue reflects the end terminal stage of the disease.

Validation of ISG15 level in blood requires further validation owing to less numbers of samples. This observation is also unique and has a potential for further study.

Overall I felt that there is a disjunction between findings of each of the component.

I think a revised version may be helpful with focus of study around ISGylation and ATXN1 pathology using mice model and human peripheral blood and other samples for its biomarker potential.

Reviewer #2 (Remarks to the Author):

This study reports on the analyses of RNA-seq data obtained from differentiating SCA1 patient-derived iPSCs to embryoid bodies and Purkinje cells that revealed the role of ISG15 to inhibit proteasome degradation of mutant ataxin-1. Importantly, the iPSC results are supported by data from a knockin mouse model of SCA1 and SCA1 patients. An intriguing finding is the elevation of ISG15 early on in plasma from SCA1 patients. As the authors note, the potential of ISG15 as disease biomarker will require additional data. The authors are commended on the extensive amount of data presented. However, extensive editing is recommended to improve English language and grammar. The current manuscript has several spots where the insufficient quality of the writing prevents comprehension and assessment. For example, on page 13 lines 410- 411 where results of autophagy and proteasome inhibitor are presented the specific inhibitors used for each pathway should be stated. On page 8, line 241 Figure 3C should be Figure 2C? There is no Figure 3C.

Reviewer #3 (Remarks to the Author):

The study “Dynamic molecular network analysis of iPSC differentiation to Purkinje cells delineates the role of ISG15 in temporal regulation of SCA1 pathology at the earliest stage” by Homma et al claims to elucidate the earliest developmental pathology of SCA1 using newly developed dynamic molecular network analyses. This is a very interesting study providing with a relevant contribution to understand the complex and unknown series of events underlying early SCA1 pathogenesis.

The manuscript is extensive and overall it is clearly written. Albeit the methodology used is thoroughly described and it seems appropriate, I have some concerns about the major conclusions proposed from the data shown.

The authors performed a challenging comprehensive RNA-seq analysis of iPSCs generated from 2 SCA1 patients, with 25/44 and 29/41 CAG repeats, being both expanded alleles at the lowest range of pathogenesis, in order to identify earlier pathological molecular events such as that originating from IL4 receptor- α (IL4R) as the causative trigger of brain inflammation reported in later stages of SCA1 that may be an early trigger for accumulation and aggregation of the disease protein (evidence for this is not provided). Presumptive supporting human evidence is the identification of higher levels of ISG15 in plasma in SCA1 albeit 9 human SCA1 patients with similar expanded CAG repeats within the range of 40 to 49 fail to show a clear correlation with either CAG size or SARA scale values. The compelling correlation is only shown between plasma ISG15 levels and disease duration. The last related paragraph on page 14 is not strongly supported by the data provided and thus should be minored and properly discussed. The missing correlation could be due to the relatively low number of expanded CAG repeats in the patients included (I assume this is because authors focus on deciphering early pathogenesis). Have the authors considered testing and including SCA1 patients with a higher number of pathological CAGs for these correlation analyses? This should be appropriately discussed in the manuscript. On the other hand, have authors studied ISG15 levels at the SCA1 pre-symptomatic stage in order to propose that ISG15 is definitively involved in very early stages of pathogenesis?

Experiments in Figure 7 in mice showing that enhanced ISGylation inhibited Ub-proteasome-dependent degradation of mutant Atxn1 in Purkinje cells could be confirmed in human SCA1 post-mortem tissues? Authors show higher levels of ISG15 expression in human SCA1 cerebellum along with the physical interaction of ISG15 and ataxin-1, thus it should be relatively easy to prove consequent enhanced ISGylation and inhibition of proteasome-dependent degradation of mutant Atxn1 in human cerebellar Purkinje cells to confirm the findings seen in mice.

IFN-dependent expression is aberrantly elevated or compromised in a myriad of human diseases, including multiple types of cancer, neurodegenerative disorders (Ataxia Telangiectasia and Amyotrophic Lateral Sclerosis), inflammatory diseases (Mendelian Susceptibility to Mycobacterial Disease (MSMD), bacteriopathy and viropathy), and in the lumbar spinal cords of veterans exposed to Traumatic Brain Injury (TBI). Thus, is not a specific trigger for a particular disease, in this case SCA1, but it seems it is a common player occurring during neurodegeneration and neuroinflammation. Strikingly, ISG15 and ISGylation are known to have both inhibitory and/or stimulatory roles in the etiology and pathogenesis of human diseases. Thus, ISG15 is considered a "double-edged sword" for human diseases in which its

expression is elevated. This needs to be properly discussed since the authors only considered the inhibitory role and not the alternative role.

Because of the roles of ISG15 and ISGylation in human diseases (nicely reviewed by Mirzalieva et al Cells 2022 but not cited in the manuscript), both ISG15 and ISGylation have been proposed to be diagnostic/prognostic biomarkers and therapeutic targets for these ailments. In this context, the sentence on page 14 “However, these results suggested a possible use of plasma ISG15 for evaluation of disease progression or of therapeutic effects of candidate drugs needs additional evidence, and I am not convinced that it is supported with the evidence provided in the study. Therefore, its conclusion tone needs to be lowered and properly discussed. Having said that, in my opinion the major caveat of the study is still to prove that the high levels of expression of ISG15 and enhanced ISGylation identified in the study are a primary result of mutant ataxin-1, thus an early trigger event, and does not reflect the neuroinflammatory process known to occur in SCA1 (see below). Again, study of ISG15 and ISGylation in SCA1 pre-symptomatic individuals could clarify whether this is a primary and early trigger in pathogenesis. Authors need to carefully consider these points in the discussion, citing at least a few recent manuscripts reviewing ISG15 and ISGylation in Human Diseases, particularly in neurodegeneration (i.e. ataxia telangiectasia which is not caused by expanded polyglutamine).

In the discussion, “ISGylation was activated in Purkinje cells of SCA1 model mice and human SCA1 patients” this should be revised since evidence of high levels of ISG15 in humans do not show unequivocal prove of ISGylation activation. Authors should change this statement of prove evidence of activation of ISGylation in humans, not only in mice.

The 475 paragraph in the discussion: “Collectively, the findings indicate that YAP and Cbln1 contribute to early-stage developmental pathology of SCA1. The result is quite surprising that SCA1 pathology initiates from such an extremely early stage of development”. This should be lowered to: “Collectively, the findings indicate that YAP and Cbln1 might contribute to early-stage developmental pathology of SCA1” since no evidence is provided that indeed it is the earliest player.

Alteration of ISG15 and ISGylation and their roles in ataxia telangiectasia should be discussed.

The sentence 506 in the discussion: “Interestingly, two recent reports suggested relevance of ISG15 to neurodegeneration or neuroinflammation”. There are studies from 1996, 2013 and 2021 noting the role of interferon beta-inducible proteins, ISG15, in ataxia telangiectasia, a neurodegenerative disorder characterized by cerebellar Purkinje cell loss like SCA1. In 1996 ISG15 and LMP2 were found to be basally elevated in ataxia telangiectasia due to constitutive activation of the IFN- β induction pathway. Notably elevated levels were detected in patients cerebella. Have the authors tested for IFN- β pathway induction in SCA1 that might result in elevated ISG15 levels? In addition to proteinopathy, aberrant activation of the ISG15 pathway also leads to mitochondriopathy. These arguments and studies should be included and properly discussed in the manuscript:

1. Siddoo-Atwal, C.; Haas, A.L.; Rosin, M.P. Elevation of interferon beta-inducible proteins in ataxia

telangiectasia cells. *Cancer Res.* 1996, 56, 443–447.

2. Juncker, M.; Kim, C.; Reed, R.; Haas, A.; Schwartzburg, J.; Desai, S. ISG15 attenuates post-translational modifications of mitofusins and congression of damaged mitochondria in Ataxia Telangiectasia cells. *Biochem. Biophys. Acta Mol. Basis Dis.* 2021, 1867, 166102.

3. Desai, S.D.; Reed, R.E.; Babu, S.; Lorio, E.A. ISG15 deregulates autophagy in genotoxin-treated Ataxia Telangiectasia cells. *J. Biol. Chem.* 2013, 288, 2388–2402.

Line 602 in the discussion “Collectively, our study reveals a new approach for unveiling the earliest pathology of neurodegenerative diseases, and discovered a previously unexpected molecule functioning at the earliest stage of developmental pathology of SCA1, which would stimulate similar approaches to the earliest pathology that may lead to prevention of neurodegeneration in the future”. Earliest should be changed for presumably early since authors could not unequivocally prove that ISG15 and ISGylation alterations are the earliest event in SCA1.

Overall, the discussion is very long and needs to be shortened without missing the relevant points supporting the conclusions. I would remove all points that not strictly supported by evidence data.

A paragraph noting that human brain samples were given with informed consent should be included in the Ethics section.

Figures:

Figure 1A: should be removed from the main figures and be included as supplementary data.

Figure 2: A, figure needs to be in a higher resolution, and larger text of pathways since it is very difficult to read.

Figure 8: ISG15 in human SCA1 brains. Authors need to indicate here estimated number (50?) of CAG repeats.

Reviewers' comments:

Reviewer #1 (Remarks to the Author):

Homma et al described a molecular study on Spinocerebellar ataxia type-1 models which includes patient derived cell lines, SCA1-Mice model and validation exercise involving peripheral blood. Overall, a voluminous work that highlights the role ISGylation of ATXN1 as marker and a process of protein modification (ATXN1 here) in SCA1 pathology. There are several caveats in the work about the presentation of various findings at each level of model system involvement. The study has three components mainly, i) generation of purkinje cell lines from lymphocytes derived iPSc, ii) RNAseq of these cell lines and iii) Atxn1 knock-in mice model for the validation.

Overall these appears somewhat connected however majorly disunited for

a) iPSC derived pan-neuronal lineage and purkinje lines: major findings of this work is at morphological level that exhibit impaired neurite length and spine density of the pan-neurons and purkinje lines. While at RNAseq the purkinje line was utilized for deriving network based molecular perturbation with connectome initiated at iPSC level. The finding of HMGB1 mediated rescue of spine density was well connected with RNAseq based network analysis.

b) Molecular Dynamic network analysis of RNAseq: Here, the temporal model of analysis was taken from iPSC lineage-Embroid body-Purkinje cells to decipher the maximally disturbed network in SCA1 model and thus extrapolated to ISGylation.

c) ISGylation model was followed in Atxn-1 knock-in mice

>>> We appreciate reviewer's understanding of the structure of our paper. We would comment that there is the fourth part for validation of ISGylation with human data.

The part (a) is the part for characterization of iPSC cells by differentiating them and verify that the mutant iPSCs actually show SCA1-like phenotype. If the reviewer understood we only used iPSC for RNA-seq, it is wrong. We performed RNAseq of iPSC, EB and Purkinje cells, then used RNAseq data at three differentiation stage data for performing the temporal model of network analysis.

The part (b) include static molecular network (which revealed TEAD, YAP and their target CBLN) (b-1) and dynamic molecular network analysis (which revealed ISGylation at the end of iPSC-EB-Purkinje analysis) (b-2).

The part (c) is verification of ISGylation by mouse samples for (b-2). The part (d), fourth part, is verification by human samples for (b-2).

We have improved the connection among these parts. Please check the edit history of the manuscript word file, in which major parts are also indicated by blue letters.

Major concerns

a) For patient derived lines component of the study

1. The fundamental point of full penetrance model of SCA1 requires uninterrupted CAG repeats of length 39-44 in human studies (<https://www.ncbi.nlm.nih.gov/books/NBK1184/>). The validity of the model per say seems less precise in terms of the repeat sizes of the PBMC were in range of 41-47 with CAT interruption. The uninterrupted length of CAG in the model is only 28-36 not at least

39-44. Moreover the repeat calculation is wrongly depicted in Fig S1A (Table, peripheral blood vs clones??)

>>> Thank you very much for pointing out this critical issue. The co-authors of this paper (Shinichi Shirai & Ichiro Yabe at Hokkaido University), who provided human SCA1 resources and information, performed Sanger sequencing again, and confirmed that SCA1A patient (46y, F) had pure CAG46 and SCA1B patient (62y, M) had pure CAG43. The repeat sequence data of these patients and of iPSCs derived from the patients were corrected (Sup Figure 1A).

2. The major drawback of the work is correlation of the data with small number of RNAseq data (n=2 patient line Vs n=1 control line) to derive the most significant expressed for the downstream analysis.

>>> Probably the reviewer looked at the table in Sup Figure 1A, and assumed that we used (SCA1A & SCA1B → n=2 patient line Vs HKC n=1 control line). But the reality was that we used SCA1A-26, SCA1B-8, SCA1B-37 for patient cell lines and 201B7, HKC-1, HKC-3 for control cell lines), meaning (n=3 patient lines Vs n=3 control lines). Though it is true that 2 patient cell lines (SCA1B-8, SCA1B-37) came from one patient (SCA1B), SCA1 is a rare disease (extremely rare in Japan and assumed to 700 people) and this was the maximum number of patients who agreed to join the research program, approved the experiments, and permitted us to generate iPSC.

The basic architecture of RNA seq data analysis was not shown, though volcano plot and pathway analysis of diff expressed genes have been presented.

>>> As requested, we added the RNAseq data analysis pipeline in Supplementary Figure 4.

No detailed list is available to decipher the key dysregulated genes in SCA1 lines vs control. Has data been deposited in SRA (NCBI)?

>>> We deposited the raw data in SRA at NCBI, and also generated a new Supplementary Data 1 for summary of dysregulated genes identified from Volcano analysis.

The raw matrix of data QC need to shown in supplementary text/table.

>>> Data QC were described in Sup Fig 4a.

3. Figure S2, line “176-79, normal and SCA1 EBs to pan-neurons, and measured neurite extension length (Supplementary Figure 2A) and density of PSD95-positive spine-like protrusions as indicators of neuronal differentiation (Supplementary Figure 2B)”. requires to be corrected.

>>> We corrected the errors in the text referring Supplementary Figure 2.

4. Line 239-242, needs to re-phrased.

>>> Figure 3C in line 241 was corrected to Figure 2C.

b) Molecular dynamic network analysis:

This seems an interesting model however in absence of through validity check of the network and its parameter in view of the temporal

nature of the limited dataset in the current work analysis the usefulness of the analysis and outcome is limited.

>>> Thank you very much. We followed the advice and improved the paper as following.

There are different sort of approach possibly require to draw major conclusion on the network analysis because this dataset is majorly a RNAseq data and in isolation will always be noisy. The first (mentioned above) component of study gave a lead around cerebellin and then luciferase reporter assay was designed and that typically shows a regulatory framework. Thus how similar framework be equally applied while the data also an temporal dynamic component.

>>> We appreciate the critical comment. We employed two approaches for network analysis (static network analysis and dynamic network analysis) from the same temporal RNA-seq data. The first approach we call static network analysis(“static” was somehow deleted by a professional English editor during editing of the former version, and made understanding of the reviewer difficult), while second approach we call dynamic network analysis.

No “cause-result relationship” was taken into account in generating the static molecular network, while “cause-result relationship” was the basis of generating the dynamic molecular network. Therefore the two approaches are not the similar framework.

In the static network analysis, significantly changed nodes were selected independently at each time point. When we tried to connect the nodes at two sequential time points, a large part of nodes were disconnected as shown in a new Supplementary Figure 8, meaning that such disconnected nodes might not influence the later

development of pathology. Therefore, we developed the dynamic molecular network (iMAD), and exclusively selected the nodes possessing the influence on the pathology at later time points.

We described in the text the details of our thoughts progressing from “static network analysis” to “dynamic network analysis”, which will unite the two components of the paper.

There are several factors which can influence the network, its edges and nodes. More robust analysis using meta-analysis of publically available dataset (of another PolyQ disease) can be brought in support of the validity (proof of principle) of this type of network analysis. The temporally dynamic data are best analyzed using Boolean networks.

>>> Thank you for the comment about suggesting Boolean networks. It is true that Boolean networks have been used in the similar cases, while Boolean networks is digital (0 or 1) and our analysis is using continuous parameters for downstream regulation.

>>> We agree that meta-analysis is a good way to verify our results. Following of the comment of the reviewer, we performed meta-analysis of publically available dataset. However, using the same SCA1 database (rather than using data of other polyQ diseases) would be more appropriate for the purpose, because if there exist a consistency between the two results from iPSCs and from public dataset in the same pathology, we can claim the validity of our method.

c) ISGylation model was followed in Atxn-1 knock-in mice: this component of work is interesting independently and owing to widely believed known roles of ISGylation in various pathologies.

Subsequently, to the mice work validation of ISG15 was performed in post-mortem brain samples.

>>> Thank you for your kind evaluation.

Line of clarification required for the observation on level of increased ISG15 level which was observed developmentally in mice, while, on the contrast post-mortem brain tissue reflects the end terminal stage of the disease.

>>> We agree with the understanding of the reviewer. However, we would like to remind the reviewer that we showed immunohistochemistry of ISG15 with mice at 56 weeks (highly aged stage), which is not so different from 60s age old at death of human patients.

Validation of ISG15 level in blood requires further validation owing to less numbers of samples. This observation is also unique and has a potential for further study.

>>> We agree with the opinion. We added three patients but this was the maximum as SCA1 is an extremely rare disease (Only 500 people in Japan and most of them are not followed by hospitals).

Overall I felt that there is a disjunction between findings of each of the component.

I think a revised version may be helpful with focus of study around ISGylation and ATXN1 pathology using mice model and human peripheral blood and other samples for its biomarker potential.

>>> We respect and appreciate the kind advice of the reviewer. We added data of peripheral blood of human patients, and also added

data on ISGylation of human mutant Atxn1 using frozen SCA1 brain sample, focusing on the human-mouse ISGylation of Atxn1.

Reviewer #2 (Remarks to the Author):

This study reports on the analyses of RNA-seq data obtained from differentiating SCA1 patient-derived iPSCs to embryoid bodies and Purkinje cells that revealed the role of ISG15 to inhibit proteasome degradation of mutant ataxin-1. Importantly, the iPSC results are supported by data from a knockin mouse model of SCA1 and SCA1 patients. An intriguing finding is the elevation of ISG15 early on in plasma from SCA1 patients. As the authors note, the potential of ISG15 as disease biomarker will require additional data. The authors are commended on the extensive amount of data presented. However, extensive editing is recommended to improve English language and grammar. The current manuscript has several spots where the insufficient quality of the writing prevents comprehension and assessment. For example, on page 13 lines 410-411 where results of autophagy and proteasome inhibitor are presented the specific inhibitors used for each pathway should be stated.

>>> We appreciate very much for kind evaluation of reviewer #2. Regarding English, we had asked professional English editing in the first submission. Also in this revision, we asked the re-editing by professional English editors.

>>> Following the advice, we added specific names of inhibitors in the text at the part pointed out by the reviewer.

On page 8, line 241 Figure 3C should be Figure 2C? There is no Figure 3C.

>>> Thank you very much. It should be Figure 2C, and we corrected the error.

Reviewer #3 (Remarks to the Author):

The study “Dynamic molecular network analysis of iPSC differentiation to Purkinje cells delineates the role of ISG15 in temporal regulation of SCA1 pathology at the earliest stage” by Homma et al claims to elucidate the earliest developmental pathology of SCA1 using newly developed dynamic molecular network analyses. This is a very interesting study providing with a relevant contribution to understand the complex and unknown series of events underlying early SCA1 pathogenesis.

>>> Thank you very much for kind evaluation.

The manuscript is extensive and overall it is clearly written. Albeit the methodology used is thoroughly described and it seems appropriate, I have some concerns about the major conclusions proposed from the data shown.

The authors performed a challenging comprehensive RNA-seq analysis of iPSCs generated from 2 SCA1 patients, with 25/44 and 29/41 CAG repeats, being both expanded alleles at the lowest range of pathogenesis, in order to identify earlier pathological molecular events such as that originating from IL4 receptor- α (IL4R) as the causative trigger of brain inflammation reported in later stages of SCA1 that may be an early trigger for accumulation and aggregation of the disease protein (evidence for this is not provided).

>>> We agree with the opinion of the reviewer. We had not performed genetic interventions of IL4R or related signal molecules to examine their later effects on accumulation and aggregation of the disease protein. These questions would be solved by mating IL-4R deficient mice with SCA1 mice. But in our idea this is beyond the scope of a

single paper, and we would bear this critical question in our mind for a next project.

Presumptive supporting human evidence is the identification of higher levels of ISG15 in plasma in SCA1 albeit 9 human SCA1 patients with similar expanded CAG repeats within the range of 40 to 49 fail to show a clear correlation with either CAG size or SARA scale values. The compelling correlation is only shown between plasma ISG15 levels and disease duration. The last related paragraph on page 14 is not strongly supported by the data provided and thus should be minored and properly discussed.

>>> Following the advice, we weakened our claim in the paragraph and carefully discussed. We added ISG15 samples from patients with the higher CAG repeat number, and could reveal the correlation between CAG size and plasma ISG15 values (Figure 9f).

The missing correlation could be due to the relatively low number of expanded CAG repeats in the patients included (I assume this is because authors focus on deciphering early pathogenesis).

>>> We appreciate understanding of the reviewer about this tissue.

Have the authors considered testing and including SCA1 patients with a higher number of pathological CAGs for these correlation analyses? This should be appropriately discussed in the manuscript. On the other hand, have authors studied ISG15 levels at the SCA1 pre-symptomatic stage in order to propose that ISG15 is definitively involved in very early stages of pathogenesis?

>>> Thank you for the advice. We searched for blood and brain samples of SCA1 patients with a higher number of CAG repeats. We could add their data, and analyzed correlation between CAG number and blood ISG15 level.

Experiments in Figure 7 in mice showing that enhanced ISGylation inhibited Ub-proteasome-dependent degradation of mutant Atxn1 in Purkinje cells could be confirmed in human SCA1 post-mortem tissues? Authors show higher levels of ISG15 expression in human SCA1 cerebellum along with the physical interaction of ISG15 and ataxin-1, thus it should be relatively easy to prove consequent enhanced ISGylation and inhibition of proteasome-dependent degradation of mutant Atxn1 in human cerebellar Purkinje cells to confirm the findings seen in mice.

>>> Thank you for the advice. We performed the immunoprecipitation with ISG15 antibody and confirmed that ISGylation was enhanced in mutant Atxn1 in human SCA1 post-mortem brain (Figure 10a). Ubiquitination was also enhanced in the precipitated ISGylated mutant Atxn1 protein (Figure 10a). Moreover, we performed cycloheximide chase assay with human iPSC-derived Purkinje cells, and confirmed that ISGylation inhibited degradation of mutant Atxn1 (Figure 10b).

IFN-dependent expression is aberrantly elevated or compromised in a myriad of human diseases, including multiple types of cancer, neurodegenerative disorders (Ataxia Telangiectasia and Amyotrophic Lateral Sclerosis), inflammatory diseases (Mendelian Susceptibility to Mycobacterial Disease (MSMD), bacteriopathy and viropathy), and in the lumbar spinal cords of veterans exposed to Traumatic Brain Injury (TBI). Thus, is not a specific trigger for a particular disease, in this case SCA1, but it seems it is a common player occurring during

neurodegeneration and neuroinflammation. Strikingly, ISG15 and ISGylation are known to have both inhibitory and/or stimulatory roles in the etiology and pathogenesis of human diseases. Thus, ISG15 is considered a "double-edged sword" for human diseases in which its expression is elevated. This needs to be properly discussed since the authors only considered the inhibitory role and not the alternative role.

>>> Thank you for the critical comment about bidirectional roles of ISG15 in the pathology. We had recognized the bidirectional roles of ISG15 and that ISG15 is involved in a wide range of diseases, and we did not claim that this is specific only to SCA1. Meanwhile, we believe, it is intriguing that our dynamic network analysis reached to ISG15 in the case of SCA1 among a numerous number of molecules involved neurodegeneration and neuroinflammation. We will discuss about both sides based on such a view.

Because of the roles of ISG15 and ISGylation in human diseases (nicely reviewed by Mirzalieva et al Cells 2022 but not cited in the manuscript), both ISG15 and ISGylation have been proposed to be diagnostic/prognostic biomarkers and therapeutic targets for these ailments. In this context, the sentence on page 14 "However, these results suggested a possible use of plasma ISG15 for evaluation of disease progression or of therapeutic effects of candidate drugs needs additional evidence, and I am not convinced that it is supported with the evidence provided in the study. Therefore, its conclusion tone needs to be lowered and properly discussed.

>>> We thank the reviewer for telling us about this interesting review paper (Mirzalieva et al Cells 2022). We had recognized ISG15 relevance to A-T while it is a disorder distinct from SCA1 in regards of causative genes and molecular mechanisms even though the main pathology is localized in the cerebellum, we had not referred them in

the previous version of our manuscript. Now, we referred this review paper (ref 72) and also original papers around this topic of A-T and ISG15.

As the reviewer said, ISG15 contribution to the A-T pathology have not been determined, while we believe that our work showed a part of it in SCA1. As the reviewer said, previous publication suggested usefulness of ISG15 as a biomarker in A-T by a model cell pathology, and our work revealed that it can be used in SCA1 in human postmortem brains and blood samples. It is possible that A-T and SCA1 share the ISG15 pathology, but it is not established of course.

So we think we can say that ISG15 may be a biomarker for evaluation of SCA1 pathology and clinical response of future drugs in SCA1, though we cannot definitely determine the role of ISG15 in the pathology of neurodegenerative diseases in general.

We modified the tone of descriptions according to this understanding.

Again, study of ISG15 and ISGylation in SCA1 pre-symptomatic individuals could clarify whether this is a primary and early trigger in pathogenesis. Authors need to carefully consider these points in the discussion, citing at least a few recent manuscripts reviewing ISG15 and ISGylation in Human Diseases, particularly in neurodegeneration (i.e. ataxia telangiectasia which is not caused by expanded polyglutamine).

>>> First, we referred the suggested papers by the reviewer.

>>> Regarding the pre-symptomatic patient samples, sampling from pre-onset patients with SCA1 gene mutations is ethically not allowed

(rejected by the ethics committee). So we discussed this kind of issue, rather than doing actual experiments with pre-onset patient samples.

In the discussion, “ISGylation was activated in Purkinje cells of SCA1 model mice and human SCA1 patients” this should be revised since evidence of high levels of ISG15 in humans do not show unequivocal prove of ISGylation activation. Authors should change this statement of prove evidence of activation of ISGylation in humans, not only in mice.

>>> We changed the sentence to “High levels of ISGylated proteins were detected in Purkinje cells of SCA1 model mice and human SCA1 patients, “. Regarding Human Purkinje cells, please see Figure 8A, 8B, 8C, 8D.

The 475 paragraph in the discussion: “Collectively, the findings indicate that YAP and Cbln1 contribute to early-stage developmental pathology of SCA1. The result is quite surprising that SCA1 pathology initiates from such an extremely early stage of development”. This should be lowered to: “Collectively, the findings indicate that YAP and Cbln1 might contribute to early-stage developmental pathology of SCA1” since no evidence is provided that indeed it is the earliest player.

>>> Regarding the following comment from the reviewer #3, we are a little bit confused. It is generally believed in developmental biology that transcription factors regulate development from ES cell stage. i.e. “developmental cascade”. This is established since my discovery of Oct-3/4 regulates ES cell differentiation as the top of cascade (Okazawa et al, Cell 1990; EMBOJ 1991, 1993), and a huge number of excellent papers from other groups support our notion, and this knowledge of Oct-3/4 directly has led to generation of iPS cells and

NOBEL Prize of Dr. Yamanaka. So, at least theoretically, the similar things can occur in any type of genetic diseases, and it is quite possible that “SCA1 pathology initiates from such an extremely early stage of development like ES cells”.

>>> However, the reviewer might claim that the non-biased finding of ours in RNA-seq might not be the fact. Anyway, we followed and changed the expression as indicated by the reviewer.

Alteration of ISG15 and ISGylation and their roles in ataxia telangiectasia should be discussed.

>>> We referred the papers related to ISG15 and ISGylation in A-T and discussed it in Discussion.

The sentence 506 in the discussion: “Interestingly, two recent reports suggested relevance of ISG15 to neurodegeneration or neuroinflammation”. There are studies from 1996, 2013 and 2021 noting the role of interferon beta-inducible proteins, ISG15, in ataxia telangiectasia, a neurodegenerative disorder characterized by cerebellar Purkinje cell loss like SCA1. In 1996 ISG15 and LMP2 were found to be basally elevated in ataxia telangiectasia due to constitutive activation of the IFN- β induction pathway. Notably elevated levels were detected in patients cerebella. Have the authors tested for IFN- β pathway induction in SCA1 that might result in elevated ISG15 levels? In addition to proteinopathy, aberrant activation of the ISG15 pathway also leads to mitochondriopathy. These arguments and studies should be included and properly discussed in the manuscript:

1. Siddoo-Atwal, C.; Haas, A.L.; Rosin, M.P. Elevation of interferon beta-inducible proteins in ataxia telangiectasia cells. *Cancer Res.* 1996, 56, 443–447.

2. Juncker, M.; Kim, C.; Reed, R.; Haas, A.; Schwartzenburg, J.; Desai, S. ISG15 attenuates post-translational modifications of mitofusins and congression of damaged mitochondria in Ataxia Telangiectasia cells. *Biochem. Biophys. Acta Mol. Basis Dis.* 2021, 1867, 166102.

3. Desai, S.D.; Reed, R.E.; Babu, S.; Lorio, E.A. ISG15 deregulates autophagy in genotoxin-treated Ataxia Telangiectasia cells. *J. Biol. Chem.* 2013, 288, 2388–2402.

>>> We performed experiments to examine IFN- β pathway induction in SCA1. As shown in new Figure 6D, FN- β and pSTAT1 were increased from the time point earlier than that of ISG15, as hypothesized by the reviewer. We appreciate the reviewer's suggestion that strengthen our story.

>>> Following the advice, we referred these papers suggested by the reviewer, and we added discussion about IFNbeta, mitochondria and autophagy in relevance to ISG15 in a new section in Discussion.

Line 602 in the discussion “Collectively, our study reveals a new approach for unveiling the earliest pathology of neurodegenerative diseases, and discovered a previously unexpected molecule functioning at the earliest stage of developmental pathology of SCA1, which would stimulate similar approaches to the earliest pathology that may lead to prevention of neurodegeneration in the future”.

Earliest should be changed for presumably early since authors could not unequivocally prove that ISG15 and ISGylation alterations are the earliest event in SCA1.

>>> This phrase is not meant for ISG15 but for whole results described in this paper in a series from cerebellin to ISG15. We changed the expression and tried to clarify what we want to say.

Overall, the discussion is very long and needs to be shortened without missing the relevant points supporting the conclusions. I would remove all points that not strictly supported by evidence data.

>>> We accepted the advice, deleted many parts and shortened Discussion.

A paragraph noting that human brain samples were given with informed consent should be included in the Ethics section.

>>> Human ethics approval was described briefly in previous “Ethics” section, but we followed the advice and changes description.

Figures:

Figure 1A: should be removed from the main figures and be included as supplementary data.

>> We followed the advice and moved previous Figure 1A to new Supplementary Figure 3.

Figure 2: A, figure needs to be in a higher resolution, and larger text of pathways since it is very difficult to read.

>> Font sizes in Figure 2A are enlarged.

Figure 8: ISG15 in human SCA1 brains. Authors need to indicate here estimated number (50?) of CAG repeats.

>> We wrote the number of 50 repeats in legend.

Reviewers' comments:

Reviewer #1 (Remarks to the Author):

All Concerns are addressed well. I have no further comments.

Reviewer #4 (Remarks to the Author):

To examine transcriptional alteration early in disease, authors differentiated patient donated iPSCs into Purkinje cells. iPSCs were created from lymphocytes donated by two SCA1 patients (one male and one female), while controls were iPSCs commercially obtained from one male and one female controls. Two clones of male control and SCA1 lines, and one clone of female control and SCA1 lines were used. After briefly characterizing PC morphology, authors performed RNA seq and used a novel method that they developed and called induction Method-based Analysis for Dynamic molecular networks (iMAD), to identify ISG15 as an important regulator of mutant ATXN1 degradation. This is an interesting study revealing novel players in SCA1 pathogenesis. However, several issues need to be addressed before publication.

1. Comparing N = 2 two biological samples for both SCA1 and control lines is borderline statistically; it is highly recommended for authors to obtain an additional SCA1 and control lines.
2. Please add following to the Methods: quantification of proliferation times of feeder-free iPSCs (Supplementary Figure 2a), quantification of the size of embryonic bodies (EBs)(Supplementary Figure 2c), and quantification of spine density (PSD95-positive) (Supplementary Figure 2e).
3. Please present the average measure per cell line and use that number to calculate statistical significance in all relevant figures (e.g. Fig .1, Supplementary Figure 2e).
4. In Fig. 1 there are no visible PC in SCA1 condition. Please include image with PCs.
5. Authors state for Fig. 1: "Eight visual fields from two independent wells were observed". Could authors please clarify how many cell lines were examined per each genotype (control vs SCA1) for data in Fig. 1?
6. Authors need to confirm that PCs are mature by showing functional evidence e.g. using ephys or calcium imaging.
7. Two of control (HKC1 and HKC3) and SCA1 (B8 and B37) lines are clones and as such not true biological replicates but more technical replicates. Could authors please clarify how was this taken into account during the RNAseq analysis?
8. In Figure 2, p values for the pathways do not seem to be statistically significant-can authors please check this and clarify?
9. In volcano plots could authors please indicate numbers of DEGs (FDR p value < 0.05)?
10. Authors state that " P-values evaluating the difference between mRNA expression of SCA1-derived cells and normal cells were calculated by Welch's t-test (N = 3) and expressed as log10 (p-value) along the y-axis." And " A Welch's test was employed to compare the difference of gene expression between SCA1-derived and normal cells. The significance level was set at 5%." Could authors please describe in more details how was RNAseq analysis done and how did they adjust/take into account multiple comparisons?
11. In Fig. 3, authors test the effect of overexpressing TEAD on luciferase constructs with different regions of Cerebelin 1 (Cbln1) regulatory elements in iPSCs lines. Could authors please rationalize this experiment and clarify the result in the view of Fig. 2 that shows that there is no difference neither in Cbln1 nor TEAD expression in control vs SCA1 iPSCs.
12. Fig. 7A-B DAPI intensity as well as background intensity for ISG15 seems to be increased in SCA1 ki mice. Can authors please check if these images were taken with the same imaging parameters (z-

stack, laser intensity, gain etc), add imaging and analysis description to methods (was z-stack used, which lobules were imaged, etc)?

13. For Fig. 7C-D and all other WB figures, please include all WB used for quantification as a supplementary figure.

14. Fig. 7F please validate results using RTqPCR.

15. In SCA1 ki mice due to additional 152 glutamate, mutant ATXN1 154Q separates from WT ATXN1 2Q on WB. However, in Fig. 8b there is no higher band of mutant ATXN1 in ki mice compared to endogenous 2Q ATXN1, thus it seems that ISG15 and Ub only target wt/endogenous ATXN1. Could authors please clarify why there is no mutant ATXN1 band in these WB?

16. In Fig. 8d, ISG15 signal in the input looks much different than in previous Figs. 8a and b -could authors clarify if there was a difference in sample preparation?

17. Authors state "The experiments supported that the 100 kDa protein was Atxn1 conjugated with both Ub and ISG15 (Figure 8d, thin blue arrow). In addition, immunoprecipitation of cerebellar tissue samples from Atxn1-KI mice, but not their normal siblings, detected both ubiquitinated and ISGylated Atxn1 at higher molecular weights (Figure 8d, thick blue arrow and blue arrowhead). The high molecular weight bands could be Atxn1 molecules cross-linked by transglutaminase 561 that escaped from protein degradation."

To increase confidence of this claim, at min authors need to repeat these experiments in ATXN1 KO mice to confirm these are indeed ATXN1 specific bands.

18. Fig. 8e could authors please clarify if green signal in top panels is ATXN1 or Ub IFC as both are stated in the figure.

19. On Figs. 8f,g and 10 please show ISG15 knockdown.

20. In Fig. 9 there are 9 SCA1 patients in the table, but there are 12 data points in the graphs-can authors please clarify what are additional 3 data points?

21. Please clarify why changes needed to be done in the commercial ELISA test and what exactly was changes and how it affected final measurements (e.g. in the calibration dilution curve). Since there is such variability in ISG15 levels in multiple sampling in patients, could authors clarify why controls were also not sampled multiple times?

22. For Figs. 10a and b please clarify how many patients and controls were used i.e. what are three dots in histograms in Fig.10b?

23. In Supplementary Fig 2 authors state "Similar experiments were repeated three times". Could they please clarify how were repeats similar and how were they different?

24. In discussion authors state: "Regarding the stage from EBs to Purkinje cells, the study revealed that Cbln1, a well-known soluble factor for cerebellum synapse elimination called synapse pruning, was an important target of TEAD-YAP transcription in iPSCs and was decreased in EBs differentiating to Purkinje cells (Figure 3). Our previous result that synaptic density is increased in the cerebellum of Atxn1-KI mice⁶⁷ is consistent with this finding. "

However, while Cbln1 is decreased, authors also found reduced spine density in SCA1 iPSCs derived neurons, which is the opposite of expected increase in synapses. Authors should discuss these possibly conflicting results.

25. Can authors clarify the last part (underlined) of this sentence in the abstract" Dynamic molecular network analysis revealed that ISG15 inhibited proteasomal degradation of mutant ataxin-1 and enhanced diseased protein accumulation in Purkinje cells of SCA1 model mice and human patients, to which molecular networks progressing from early embryonic stage finally reach after birth."

Reviewers' comments:

Reviewer #1 (Remarks to the Author):

All Concerns are addressed well. I have no further comments.

Reviewer #4 (Remarks to the Author):

To examine transcriptional alteration early in disease, authors differentiated patient donated iPSCs into Purkinje cells. iPSCs were created from lymphocytes donated by two SCA1 patients (one male and one female), while controls were iPSCs commercially obtained from one male and one female controls. Two clones of male control and SCA1 lines, and one clone of female control and SCA1 lines were used. After briefly characterizing PC morphology, authors performed RNA seq and used a novel method that they developed and called induction Method-based Analysis for Dynamic molecular networks (iMAD), to identify ISG15 as an important regulator of mutant ATXN1 degradation. This is an interesting study revealing novel players in SCA1 pathogenesis. However, several issues need to be addressed before publication.

>>> We thank the reviewer #4 for evaluation. But we have asked the editor about our stance to respond to reviewer #4, because this is not the first review and reviewer #4 was simply asked to play the role of reviewer #3 but not a new reviewer and because reviewer #4 inquired us many new experiments.

The editors has answered to us that we only need to respond to comment 14 with an experiment.

1. Comparing N = 2 two biological samples for both SCA1 and control lines is borderline statistically; it is highly recommended for authors to obtain an additional SCA1 and control lines.

>>> We received the same comment from reviewer #1 in the previous review and have already responded to it.

2. Please add following to the Methods: quantification of proliferation times of feeder-free iPSCs (Supplementary Figure 2a), quantification of the size of embryonic bodies (EBs)(Supplementary Figure 2c), and quantification of spine density (PSD95-positive) (Supplementary Figure 2e).

>>> We improved the descriptions in the methods following the comments.

3. Please present the average measure per cell line and use that number to calculate statistical significance in all relevant figures (e.g. Fig .1, Supplementary Figure 2e).

>>>> We received such indications for presentation from other reviewer in the previous round, and we have changed our description following the previous indications.

4. In Fig. 1 there are no visible PC in SCA1 condition. Please include image with PCs.

>>> We used another image with PCs.

5. Authors state for Fig. 1: "Eight visual fields from two independent wells were observed". Could authors please clarify how many cell lines were examined per each genotype (control vs SCA1) for data in Fig. 1?

>>> We used two lines in the previous version. Assuming the intention of reviewer 4 in this question, we added one line. So in this revised version, we used three lines.

>>> We described numbers of cell lines (N) and numbers of wells (n) in Figure 1.

6. Authors need to confirm that PCs are mature by showing functional evidence e.g. using ephys or calcium imaging.

>>> This is a new inquiry. In addition the original paper describing the method has shown such data.

7. Two of control (HKC1 and HKC3) and SCA1 (B8 and B37) lines are clones and as such not true biological replicates but more technical replicates. Could authors please clarify how was this taken into account during the RNAseq analysis?

>>> The same as our response to comment 1, we have already answered.

8. In Figure 2, p values for the pathways do not seem to be statistically significant-can authors please check this and clarify?

>>> Adjusted p-values after B-H procedure were above 0.05 in most pathways, as the reviewer commented.

The panels in Figure 2 simply showed top 15 pathways in statistical significance, as we wrote in the figure legend == Right panels show KEGG pathway analysis of the changed genes (Results of pathways with the top 15 smallest p-values are shown) ==.

Therefore in this revision, we added description in the text as follows.

“Fisher’s exact test indicated that 14, 33 and 3 pathways were significantly affected ($p < 0.05$) at iPSC, EB and Purkinje cell stage, respectively, while only top five pathways at EB stage kept their statistical significances ($p < 0.05$) after multiple testing correction (Benjamini-Hochberg procedure) among all KEGG pathways (Figure 2a)”.

The details of statistical significance in these KEGG pathways are added to the source data (Supplementary Data 5).

9. In volcano plots could authors please indicate numbers of DEGs (FDR p value < 0.05)?

>>> We described numbers of DEGs (uncorrected p-value < 0.05) in Figure 2 legend.

10. Authors state that “ P-values evaluating the difference between mRNA expression of SCA1-derived cells and normal cells were calculated by Welch’s t-test (N = 3) and expressed as log₁₀ (p-value) along the y-axis.” And “ A Welch’s test was employed to compare the difference of gene expression between SCA1-derived and normal cells. The significance level was set at 5%.” Could authors please describe in more details how was RNAseq analysis done and how did they adjust/take into account multiple comparisons?

>>> We did not correct raw p-value by Welch’s t-test, and added this information to the method.

11. In Fig. 3, authors test the effect of overexpressing TEAD on Luciferase constructs with different regions of Cerebelin 1 (Cbln1) regulatory elements in iPSCs lines. Could authors please rationalize this experiment and clarify the result in the view of Fig. 2 that shows that there is no difference neither in Cbln1 nor TEAD expression in control vs SCA1 iPSCs.

>>> We had described the rational for using iPSC in Luciferase assay in the text section for Figure 2, and previous round reviewers including Reviewer #3 were satisfied with our explanation of the rational.

12. Fig. 7A-B DAPI intensity as well as background intensity for ISG15 seems to be increased in SCA1 ki mice. Can authors please check if these images were taken with the same imaging parameters (z-stack, laser intensity, gain etc), add imaging and analysis description to methods (was z-stack used, which lobules were imaged, etc)?

>>> We used the same imaging parameters but auto-correction function of the microscopy might have changed the signals. We also described the requested information (z-stack, which lobule) in methods.

13. For Fig. 7C-D and all other WB figures, please include all WB used for quantification as a supplementary figure.

>>> It is a new and exceptionally unusual inquiry unnecessary to respond considering the role of reviewer #4. We also think this is out of rule. We consulted with editors and decided not to respond.

14. Fig. 7F please validate results using RTqPCR.

>>> We performed RT-PCR to validate the results in Fig 7F.

15. In SCA1 ki mice due to additional 152 glutamate, mutant ATXN1 154Q separates from WT ATXN1 2Q on WB. However, in Fig. 8b there is no higher band of mutant ATXN1 in ki mice compared to endogenous 2Q ATXN1, thus it seems that ISG15 and Ub only target wt/endogenous ATXN1. Could authors please clarify why there is no mutant ATXN1 band in these WB?

>>> We think the reviewer is saying glutamine instead of glutamate.

>>> It is related to comment #16. As the reviewer might know, if he/she is a researcher of polyQ or degenerative diseases, the band is detected at a much higher position than the calculated MW size, and it differs in different gels. We can see the higher bands of mutant and normal (now indicated with bold arrows and arrowheads) Atxn1 in Fig 8b, and they were ISGylated.

16. In Fig. 8d, ISG15 signal in the input looks much different than in previous Figs. 8a and b -could authors clarify if there was a difference in sample preparation?

>>> In Fig 8a and b, we used RIPA buffer for extraction, while we used TNE buffer in Fig 8d. This might cause the difference in additional bands.

17. Authors state “The experiments supported that the 100 kDa protein was Atxn1 conjugated with both Ub and ISG15 (Figure 8d, thin blue arrow). In addition, immunoprecipitation of cerebellar tissue samples from Atxn1-KI mice, but not their normal siblings, detected both ubiquitinated and ISGylated Atxn1 at higher molecular weights (Figure 8d, thick blue arrow and blue arrowhead). The high molecular weight bands could be Atxn1 molecules cross-linked by transglutaminase 561 that escaped from protein degradation.”

To increase confidence of this claim, at min authors need to repeat these experiments in ATXN1 KO mice to confirm these are indeed ATXN1 specific bands.

>>> This is a new inquiry (KO mice). The editor has judged new experiments are not necessary in this case.

18. Fig. 8e could authors please clarify if green signal in top panels is ATXN1 or Ub IFC as both are stated in the figure.

>>> Green signals were Atxn1 in upper panels, and this was an error of us.

19. On Figs. 8f,g and 10 please show ISG15 knockdown.

>>> This is a new request from the reviewer 4, which is judged by the editor unnecessary to do. However, we added the data of ISG15 blots in Fig 8f, g and 10.

20. In Fig. 9 there are 9 SCA1 patients in the table, but there are 12 data points in the graphs-can authors please clarify what are additional 3 data points?

>>> In the previous review, we received the comment suggesting to increase N of samples. Therefore, we added 3 normal and 3 SCA1 samples. We changed the graphs but forgot to change the numbers below graphs and the information of the table. We now corrected these errors.

21. Please clarify why changes needed to be done in the commercial ELISA test and what exactly was changes and how it affected final measurements (e.g. in the calibration dilution curve). Since there is such variability in ISG15 levels in multiple sampling in patients, could authors clarify why controls were also not sampled multiple times?

>>> What exactly changed was already written in details in the text (page 15) and in the Methods of previous version R1. And the reason why changes were necessary was needless to say, because we measured “human ISG15” but the ELISA kit was for “mouse ISG15”. Calibration dilution curve after changing the secondary antibody was shown in Supplementary Data 5, in order to confirm soundness of the modified ELISA.

In the latter half of the reviewer’s comment, the reviewer #4 seems to accuse that normal samples were not taken at multiple time points. This is a retrospective study but not a prospective study. Hence, people who are diagnosed as normal will never come again to the hospital.

22. For Figs. 10a and b please clarify how many patients and controls were used i.e. what are three dots in histograms in Fig.10b?

>>> N= 3 brains from patients and 3 normal people. We wrote in the figure.

23. In Supplementary Fig 2 authors state “Similar experiments were repeated three times”. Could they please clarify how were repeats similar and how were they different?

>>> This is actually new inquiry that reviewer #3 did not ask, so we think it is unnecessary to answer. But we added graphs to source data (Supplementary Data 5), so the reviewer #4 may see how they are similar or different (statistically subgroups were not different, and we do not think it unnecessary to show the new statistics).

24. In discussion authors state: “Regarding the stage from EBs to Purkinje cells, the study revealed that Cbln1, a well-known soluble factor for cerebellum synapse elimination called synapse pruning, was an important target of TEAD-YAP transcription in iPSCs and was decreased in EBs differentiating to Purkinje cells (Figure 3). Our previous result that synaptic density is increased in the cerebellum of Atxn1-KI mice⁶⁷ is consistent with this finding. “ However, while Cbln1 is decreased, authors also found reduced spine density in SCA1 iPSCs derived neurons, which is the opposite of expected increase in synapses. Authors should discuss these possibly conflicting results.

>>> This is a new inquiry. We have already discussed the style of discussion in the previous review. In brief, iPSC-derived neurons are pan-neurons not PCs. And Cbln1 is changed in the extremely early stage (EB). The similar molecular change could lead later to different phenotypes in different cell types in general. Readers can easily understand the reason without such explanations.

25. Can authors clarify the last part (underlined) of this sentence in the abstract” Dynamic molecular network analysis revealed that ISG15 inhibited proteasomal degradation of mutant ataxin-1 and enhanced diseased protein accumulation in Purkinje cells of SCA1 model mice and human patients, to which molecular networks progressing from early embryonic stage finally reach after birth.”

>>> “Underline” was disappeared when we received the comment from editorial office. We believe the meaning of this sentence is clear.

Reviewers' comments:

Reviewer #4 (Remarks to the Author):

Authors were instructed by editor to respond only to comment 14 from the initial evaluation. Authors well addressed this comment by performing and including RTqPCR results in Fig 7g.

Minor suggestion for Fig 7g is to either indicate what a.u. on the y axes stands for, or if it means absolute units remove a.u. from these graphs as the results are not absolute units but are normalized to Gapdh and state on y axes that the results were normalized using Gapdh, and add if iPSC stage was used as reference in the calculations. Additionally, describing in the legend if authors used independent RNA samples or the same samples that they used for RNAseq would increase reproducibility.

Additional minor suggestions are:

1. In the result portion of the manuscript authors state "Patient A carried 25/44 repeats, and patient B carried 29/41 repeats, as demonstrated by automated fragment analysis of lymphocytes and Sanger sequencing confirming the corresponding (CAG)_mCAT(CAG)_n sequences (Supplementary Figure 1a)." This should be made consistent with data shown in Supp Fig 1a that shows that patient A has 26/46 repeats in peripheral blood and 26/46 and 26/47 in cell lines, and patient B 29/43 in peripheral blood and 29/43 and 29/44, 29/46, 29/47 in derived cell lines.

2. In results description of fig 10 states " To confirm the findings obtained in mice, we examined enhanced ISGylation and inhibition of proteasomal degradation of human mutant Atxn1 in cerebellar tissues of a post-mortem patient (28/53 CAG repeats) (Figure 10)" which indicates that tissue from one patient was analyzed.

As Fig 10b uses brain tissues from N=3 SCA1 patients, authors should change text to reflect this. Also Fig 10 c and d are mentioned as 10 a and b in text.

3. In their response authors state "It is related to comment #16. As the reviewer might know, if he/she is a researcher of polyQ or degenerative diseases, the band is detected at a much higher position than the calculated MW size, and it differs in different gels. We can see the higher bands of mutant and normal (now indicated with bold arrows and arrowheads) Atxn1 in Fig 8b, and they were ISGylated. "

It is great that authors included the bold arrow and arrowhead to show where mutant and normal ATXN1 should be, however it is really hard to see bands. I suggest more representative image where bands are clearly visible.

Reviewers' comments:

Reviewer #4 (Remarks to the Author):

Authors were instructed by editor to respond only to comment 14 from the initial evaluation. Authors well addressed this comment by performing and including RTqPCR results in Fig 7g.

Minor suggestion for Fig 7g is to either indicate what a.u. on the y axes stands for, or if it means absolute units remove a.u. from these graphs as the results are not absolute units but are normalized to Gapdh and state on y axes that the results were normalized using Gapdh, and add if iPSC stage was used as reference in the calculations.

>>> Thank you very much for pointing out our elementary mistake. Fig 7g did not and does not exist, and we think that the reviewer's comment meant Fig 7f right panel. Here the value of y-axis means exactly what the reviewer said as the latter case. We deleted a.u., changed the indication of Y-axis from "corrected signal" to "corrected CT value", and added its explanation in Figure legend.

Additionally, describing in the legend if authors used independent RNA samples or the same samples that they used for RNAseq would increase reproducibility.

>>> We used independent RNA samples, and described it in the figure 7f legend.

Additional minor suggestions are:

1. In the result portion of the manuscript authors state “Patient A carried 25/44 repeats, and patient B carried 29/41 repeats, as demonstrated by automated fragment analysis of lymphocytes and Sanger sequencing confirming the corresponding (CAG)_mCAT(CAG)_n sequences (Supplementary Figure 1a).”

This should be made consistent with data shown in Supp Fig 1a that shows that patient A has 26/46 repeats in peripheral blood and 26/46 and 26/47 in cell lines, and patient B 29/43 in peripheral blood and 29/43 and 29/44, 29/46, 29/47 in derived cell lines.

>>> Thank you very much for pointing out the confusing mismatches of numbers.

In the first round of review, reviewer 1 pointed out our errors in Sanger sequencing, and to which we repeated Sanger sequence (co-authors of Hokkaido University) and corrected the second column of mutant allele sequences in the lower Table of Sup Fig 1a.

We indicated the repeat number analyzed by sequencing in the third column of Table, while the indication of the third column remained as Fragment analysis. Therefore, this comment of the reviewer means the number based on Sanger sequencing. Anyway we appreciate very much the comment of the reviewer, by which we could exclude our errors.

We corrected these mismatches by changing the Sup Fig 1a, the result description for Sup Figure 1a and also the legend for Sup Figure 1a. In new Sup Fig 1a, the upper panel shows the results of fragment analysis, while the lower Table only shows the results of Sanger sequencing. We think this change will simplify the presentation and exclude the confusion. Accordingly we changed the description in the text and legend.

2. In results description of fig 10 states “ To confirm the findings obtained in mice, we examined enhanced ISGylation and inhibition of proteasomal degradation of human mutant Atxn1 in cerebellar tissues of a post-mortem patient (28/53 CAG repeats) (Figure 10)” which indicates that tissue from one patient was analyzed.

As Fig 10b uses brain tissues from N=3 SCA1 patients, authors should change text to reflect this. Also Fig 10 c and d are mentioned as 10 a and b in text.

>>> Before answering the first part of the reviewer’s comment, we apologize that, in the last paragraph of Result section, indications of Figure 10 panels were not correct. We corrected these errors, so please check the text descriptions. This is the answer for the second part of the reviewer’s comment.

In Figure 10a & b, we used postmortem brains as the sample, while in Figure 10c & d, we used iPS cell line derived neurons.

In the latter case, we used N=3 cell lines, while in the former case, we could obtain only one sample of frozen postmortem brain tissue from Dr. Saito at Tokyo Metropolitan Institute of Gerontology. We also asked the co-authors, Drs. Yabe and Shirai, in Hokkaido University which has been following the largest number of SCA1 patients in Japan, but they answered that they do not have frozen brain tissues of SCA1.

In immunohistochemistry, meanwhile, we could use paraffin sections derived from three SCA1 patients (so N=3).

3. In their response authors state “It is related to comment #16. As the reviewer might know, if he/she is a researcher of polyQ or degenerative diseases, the band is detected at a much higher position than the calculated MW size, and it differs in different gels.

We can see the higher bands of mutant and normal (now indicated with bold arrows and arrowheads) Atxn1 in Fig 8b, and they were ISGylated. “ It is great that authors included the bold arrow and arrowhead to show where mutant and normal ATXN1 should be, however it is really hard to see bands. I suggest more representative image where bands are clearly visible.

>>> We replaced the image with another one made by a longer time exposure, in which the bands can be recognized obviously. We also added magnified images of Atxn1 bands in right panels of Fig 8b.

** See the Nature Portfolio author and referees' website at www.nature.com/authors for information about policies, services and author benefits

Communications Biology is committed to improving transparency in authorship. As part of our efforts in this direction, we are now requesting that all authors identified as ‘corresponding author’ create and link their Open Researcher and Contributor Identifier (ORCID) with their account on the Manuscript Tracking System prior to acceptance. ORCID helps the scientific community achieve unambiguous attribution of all scholarly contributions. You can create and link your ORCID from the home page of the Manuscript Tracking System by clicking on ‘Modify my Springer Nature account’ and following the instructions in the link below. Please also inform all co-authors that they can add their ORCIDs to their accounts and that they must do so prior to acceptance.

For more information please visit

<http://www.springernature.com/orcid>

If you experience problems in linking your ORCID, please contact the Platform Support Helpdesk.

This email has been sent through the Springer Nature Tracking System NY-610A-NPG&MTS

Confidentiality Statement: This e-mail is confidential and subject to copyright. Any unauthorised use or disclosure of its contents is prohibited. If you have received this email in error please notify our Manuscript Tracking System Helpdesk team at <http://platformsupport.nature.com> .

Details of the confidentiality and pre-publicity policy may be found here <http://www.nature.com/authors/policies/confidentiality.html>
Privacy Policy | Update Profile